# Impacts of Industrial Modification on the Structure and Gel Features of Soy Protein Isolate and its Composite Gel with Myofibrillar Protein

**DOI:** 10.3390/foods12101982

**Published:** 2023-05-13

**Authors:** Zhaodong Hu, Yichang Wang, Zihan Ma, Tianfu Cheng, Zengwang Guo, Linyi Zhou, Zhongjiang Wang

**Affiliations:** 1College of Food Science, Northeast Agricultural University, Harbin 150030, China; nodanger2021@163.com (Z.H.); gzwname@163.com (Z.G.); 2College of Food and Health, Beijing Technology and Business University, Beijing 100048, China

**Keywords:** soybean protein isolate, industrial modification methods, myofibril protein, structure, gel properties

## Abstract

Native soy protein isolate (N-SPI) has a low denaturation point and low solubility, limiting its industrial application. The influence of different industrial modification methods (heat (H), alkaline (A), glycosylation (G), and oxidation (O)) on the structure of SPI, the properties of the gel, and the gel properties of soy protein isolate (SPI) in myofibril protein (MP) was evaluated. The study found that four industrial modifications did not influence the subunit composition of SPI. However, the four industrial modifications altered SPI’s secondary structure and disulfide bond conformation content. A-SPI exhibits the highest surface hydrophobicity and I_850/830_ ratio but the lowest thermal stability. G-SPI exhibits the highest disulfide bond content and the best gel properties. Compared with MP gel, the addition of H-SPI, A-SPI, G-SPI, and O-SPI components significantly improved the properties of the gel. Additionally, MP-ASPI gel exhibits the best properties and microstructure. Overall, the four industrial modification effects may impact SPI’s structure and gel properties in different ways. A-SPI could be a potential functionality-enhanced soy protein ingredient in comminuted meat products. The present study results will provide a theoretical basis for the industrialized production of SPI.

## 1. Introduction

Soy protein isolate (SPI) is a vegetable protein with high nutritional value. It is widely used in the food industry due to its balanced composition of basic amino acids, desirable functional characteristics, and low cost [1]. SPI is a mixture of various proteins, and the main components of SPI are glycinin (11S protein) and β-conglycinin (7S protein). The 7S and 11S globulins have a globular conformation, which causes poor functional properties [2]. These functional properties significantly reduce its effective utilization [3]. To further promote the application of SPI, extensive studies on techniques such as heat treatment, high pressure, glycosylation, and changing pH and ion strength have been applied to enhance the functional properties of SPI [4,5,6,7]. The functional properties of the SPI produced depend on the processing conditions; the most common factors include heat, alkaline modification, glycosylation, oxidation, protein concentration, and other factors. In addition, modification methods include heat, alkaline modification, glycosylation, and oxidation, which are more suited for large-scale production and facilitation production in the food industry. Heat processing modulates the protein structure by denaturation and aggregation [8]. Glycosylation is a covalent interaction between the ε-amino groups of the protein molecules and the terminal, which reduces the carbonyl groups of saccharide molecules via the Maillard reaction. This procedure has been confirmed to be a relatively safe, simple, and promising method to improve the functional properties of proteins [9]. pH-shifting modifies protein structures by exposing proteins to extreme pH levels [10]. Protein is exposed to extreme pH conditions, inducing its structural unfolding and refolds, and confers enhanced physicochemical properties to protein molecules. Dong et al. found that alkaline pH-shifting was more effective for the structural modification of proteins [11]. Oxidation leads to significant changes in SPI structure, such as the modification of amino acid side chains, protein unfolding, and protein crosslinking/aggregation, which can affect SPI’s physicochemical and functional properties [4,12]. However, Current research on modification methods mainly focuses on laboratory aspects, and large differences between the laboratory and industry results remain. There is almost no research on SPI modification methods in industrial production processes. Therefore, the research on SPI industrial modification methods seems to be an imperious need.

SPI is frequently used in the food industry because of its ability to form a gel with desirable sensory and physicochemical characteristics. The ability to form gel is the basis for traditional Asian soy products, such as tofu. In addition, SPI is often applied to improve the texture of sausage. Replacing meat protein in food formulations with other sources has attracted growing interest [13]. SPI has been widely used as a nonmeat protein additive in the meat processing industry based on gel properties [14]. In particular, some previous studies have demonstrated the potential of SPI as an additive in meat products; SPI could be utilized as a component to form a solidified matrix, prevent fat and juice migration to the surface, and improve overall taste [15]. Akesowan et al. found that adding soy protein to pork sausage improved its textural properties, including its hardness, chewiness, and cohesiveness [16]. Therefore, improving the gelation features of SPI and utilizing it in meat product processing to enhance the quality of the product is a research area of great concern. Moderate oxidation improved the binding between SPI and MP, increasing the springiness of the mixed protein system [17]. Preheating soy protein boosted MP’s elastic modulus and gel intensity, although N-SPI without denaturation had a negative disruptive effect [18]. Jiang et al. found that extremely acidic conditions (pH 1.5) significantly enhanced the gelation capacity of MP in SPI, but that extremely alkaline conditions (pH 12) had no substantial impact on the gel features of MP in SPI [19]. Many studies have reported that thermal modification, alkaline modification, glycosylation, and oxidative treatment can all affect the functional characteristics of SPI and improve the potential of SPI as an additive in meat products. However, there have been few studies on industrial modifications to SPI to improve its functional properties and the potential of SPI as an additive in meat products.

Therefore, evaluating the impacts of SPI structure and the properties of the gel resulting from the four industrial modifications, as well as the gel properties of SPI-MP composites, is essential. This study explains the association between SPI structural changes and gel features and investigates the gel properties of the SPI-MP composite. The findings of this study provide a comprehensive set of data for industrialized modified SPI improvement and can be compared with laboratory data. This work further reveals the functional properties of industrialized modified SPI and the potential of SPI as an additive in meat products, which will provide process parameters and conditions for materials used for producing SPI with high industrial value. It will also provide a theoretical basis for further development and application of soy protein products.

## 2. Materials and Methods

### 2.1. Materials

Native SPI, industrial heat-modified SPI, industrial alkali-modified SPI, industrial glycosylation-modified SPI, and industrial oxidation-modified SPI were all prepared in Shandong Wonderful Industrial Group Co., Ltd. (Dezhou, China). In addition, pork longissimus muscle (24 h postmortem) was acquired from a profitable local Walmart store (Harbin, China), transported to the laboratory, and used on the same day. All of the substances and components were of analytical degree.

### 2.2. Formulation of N-SPI, A-SPI, G-SPI, and O-SPI

The industrialized invention process of SPI was performed according to the assay described by Campbell et al. [20]. As shown in Figure 1, four industrial modification methods were used, which were selected based on repeated experiments and industry screening, and all of the parameters used were optimized for each method. The abbreviations used for each protein isolate are as follows: SPI (N-SPI), industrial heat-modified SPI (H-SPI), industrial alkali-modified SPI (A-SPI), industrial glycosylated SPI (G-SPI), and industrial oxidative SPI (O-SPI).

The procedures were modified as follows:

① H-SPI: The flash temperature in the flash evaporation step was adjusted from 72 °C to 81 °C, and the other parameters remained unchanged.

② A-SPI: The pH of the neutralization sample was amended from 7.3 to 8.7, and the other parameters remained unchanged.

③ G-SPI: 1% konjac starch was added during formulation, and the mixture was heated and stirred to 75 °C for 90 min. The other parameters remained unchanged.

④ O-SPI: The SPI was placed in a 45 °C storage warehouse for one week for oxidation, and the other parameters remained unchanged.

### 2.3. SDS-PAGE

Electropherograms of industrially modified SPI on sodium dodecyl sulfate-polyacrylamide gels were obtained using the assay described in Laemmli et al. [21]. Briefly, 15% separation and 5% stacking gel were used. The protein was diluted to 1 mg/mL using a sample buffer, heated for 5 min at 95 °C, and 10 μL of the prepared sample was loaded onto the SDS-PAGE gels. The voltage was set to 120 V and 80 V in the separating and stacking gels, respectively. Imaging was performed after overnight destaining with Coomassie brilliant blue R-250.

### 2.4. Raman Spectroscopy

Following the procedures of Xiao et al. [22], with a few minor adjustments, the structural changes of SPI following industrial modification were assessed using a Raman spectrometer (HR 800, Horiba Jobin Yvon, Palaiseau). The final spectra were an average of five scans, and the spectra were read in the 600–1800 cm^–1^ range with a 4.0 cm^–1^ spectral resolution and 30 s of exposure time. Using Lab spec software, individual band intensities were smoothed, adjusted for baseline, and standardized using the phenylalanine band at 1004 cm^−1^.

### 2.5. Surface Hydrophobicity (H_0_) Measurement

Using an ANS approach with minor alterations (Kato and Nakai) [23], the Ho of the samples was determined. Sample solutions were thinned to a 0.01–0.2 mg/mL protein dose in deionized water (pH 7.0). Each thinned solution (4 mL) was then given 20 μL of the ANS (mM) reagent, which was then left at RT for 15 min. A fluorescence spectrophotometer was utilized to measure the fluorescence intensity. The emission wavelength was 470 nm, while the excitation wavelength was 390 nm (slit width 5 nm). The initial slope of the fluorescence intensity plotted versus the associated protein dose yielded the Ho value.

### 2.6. Free Sulfhydryl Content Determination

Using Wang’s method [24] with a few minor adjustments, the free sulfhydryl quantity of the SPI after industrial modification was determined. The 0.1 M phosphate buffer (pH 7.0), which contained 4 mM EDTA, 0.086 M Tris, and 0.09 M glycine, was used to create sample solutions (7.5 mg/mL), and then 67 μL of Ellman’s reagent was added. The resultant suspension liquid underwent a 1 h reaction at ambient temperature. Finally, the suspension liquid was centrifuged for 30 min (10,000× *g*). We estimated the values using a molar extinction coefficient of 13,600 M^−1^ cm^−1^ and a supernatant absorbance at 412 nm.

### 2.7. Differential Scanning Calorimetry Analysis

Tang’s assay was followed to determine the thermal stability of the SPI structure using a differential scanning calorimeter (Netzsch, Selb, Germany). Five milligrams of SPI samples with different industrial modification treatments were weighed, blended with 10 μL of 0.01 mol/L phosphate buffer (pH 7.0), placed in an aluminum plate, sealed, and kept at RT for 8 h. The aluminum pan containing the sample was located on the left edge of the DSC instrument console, while the blank aluminum pan was positioned on the right edge. The temperature scanning scale was 20~110 °C, and the heating ratio was 10 °C min^−1^. Pyris 6.0 software was used to acquire and process the data, resulting in the DSC curve of SPI. SPI’s denaturation temperature (T_D_) was obtained from the peak temperature, and the denaturation enthalpy change (ΔH) was computed from the area between the curve and the baseline.

### 2.8. Dynamic Oscillatory Rheology

SPI samples were dissolved in H_2_O, magnetically stirred for 4 h at RT, and centrifugated at 10,000× *g* for 30 min. The protein concentration in the upper phase was measured by the biuret assay, and the protein dose in the supernatant was diluted to 10% (*w*/*w*). A rotational rheometer (Model Bohlin VOR Bohlin Instruments, Inc., Cranbury, NJ, USA) was used for the measurement after vacuum degassing. The rheometer dish with a width of 40 mm and 0° was selected, the probe size was 20 mm, and the plate spacing was 1 mm. The plates were filled with a protein solution, and the excess solution was wiped off. Small drophosphate buffers of silicone oil were attached to the top of the bowl, and a protective shield was placed to avoid water evaporation.

#### 2.8.1. Temperature Sweep

The initial temperature was 20 °C, then raised to 95 °C at 2 °C/min. The temperature was held at 95 °C for 30 min, then reduced to 20 °C at 2 °C/min. The angular frequency was set to 0.63 rad/s throughout the process, and the maximum strain was 1%. The parameters such as G′ and G″ in the whole process were measured.

#### 2.8.2. Frequency Sweep

After the temperature sweep, the temperature was held at 20 °C, the maximum strain was 1%, and the sweep frequency was 0.1~10 Hz. The parameters such as G′ and G″ in the whole process were measured.

#### 2.8.3. Dynamic Rheological of MP and MP-SPI Composite Gels

Gels were created by heating from 20 to 80 °C at a rate of 1 °C/min. Each sample was repeatedly sheared in an oscillatory mode with a maximum strain of 0.02 at a fixed frequency of 0.1 Hz. For the rheological characterization of the gels, the storage modulus (G′) and loss modulus (G″) were quantified.

### 2.9. Formulation of MP

Xia, Kong, Liu, and Liu [25] used the procedure to prepare MP. The procedure was performed in a four-degree chilling chamber. Using bovine serum albumin as a reference, the biuret technique was used to determine protein concentration [26].

#### 2.9.1. Formulation of Blended Protein Mixtures

MP extract with a protein content of about 3% was diluted to 40 mg/mL. SPI (N-SPI, H-SPI, A-SPI, G-SPI, O-SPI) was prepared at 40 mg/mL (*w*/*v*) using distilled water in a 40 mM phosphate buffer that included 0.6 M NaCl (pH 6.25). MP and SPI were mixed in a 4:1 ratio and homogenized to achieve a 40 mg/mL protein content and adjusted to pH 7.0 in the well-mixed composite protein solution.

#### 2.9.2. Preparation of SPI-MP Composite Gel

A total of 20 mL of well-mixed composite protein solution was heated in a water bath at 80 °C for 30 min, then quickly cooled in ice water and stored in a 4 °C refrigerator for 12 h. The prepared gel was allowed to reach equilibrium at room temperature (20~25 °C) for 30 min before each measurement. For convenience, the composite gels prepared from MP with added N-SPI, H-SPI, G-SPI, and O-SPI are abbreviated as MP-NSPI, MP-HSPI, MP-ASPI, MP-GSPI, and MP-OSPI, respectively.

#### 2.9.3. Textural Profile Analysis (TPA)

A texture profile analysis (TPA) of the SPI-MP gels was performed using a texture analyzer (TA-XT plus, Instruments Ltd., USA) equipped with probe P/36R (3.6 cm diameter), as described by (Mi et al., 2021) with minor modifications. The gel samples were cut into cylinders (diameter, 10 mm; height, 10 mm), equilibrated for 2 h at room temperature, and then compressed at a compression degree of 50%. The pre-test and post-test speeds were set at 3 mm/s, and the test speed was set at 0.3 mm/s to obtain the TPA parameters (hardness, chewiness, springiness, and cohesiveness). The trigger type was set to auto with a 5.0 g trigger force. All samples are measured in 6–8 parallel.

#### 2.9.4. The WHC of Gel

The WHC (water holding capacity) (%) of MP gel was determined via the assay explained by Niu, et al. [27]. The findings were conveyed as a proportion (*w*/*w*) of H_2_O maintained in the gel after centrifugation (10,000× *g*, 15 min, 4 °C), using the following equation:$$\mathrm{WHC}\;(\%)=(1\;-\;\frac{W_{0}-W_{1}}{W_{0}})\times 100$$
where W_0_ and W_1_ represent the weight (g) of the gel before centrifugation and the weight (g) of the gel after centrifugation, respectively.

#### 2.9.5. Microstructure of the Gel

The process described by Haga and Ohashi [28] was modified somewhat to study the microstructure of the mixed gels via a Hitachi-S-3400N field emission SEM (Hitachi High Techno., Corp., Tokyo, Japan). First, gel samples (3 × 3 × 3 mm^3^) created via the technique above were placed, for at least 1.5 h, in a 2.5% glutaraldehyde 0.1 M phosphate buffer (pH 7.0). Then, after being washed three times with the same buffer for 10 min, the gel samples were dehydrated by being dipped in a series of solutions (50, 70, 80, 90, and 100%) of ethanol for 10 min per solution. Finally, each sample was lyophilized, sputter-coated with gold, and observed at an acceleration voltage of 10 kV and a magnification of 1000. In the SEM observations, there were three replications. For each replica, six pictures were taken.

### 2.10. Statistical Analysis

The results were provided as mean ± SD for every experiment conducted at least in triplicate. Data were processed using one-way ANOVA and Duncan’s test in the SPSS statistical package (SPSS 16.0, IBM, New York, NY, USA). The statistical significance between the two means was determined at the 95% confidence level.

## 3. Results

### 3.1. Effects of Industrial Modification Treatment on SDS-PAGE of SPI

SDS-PAGE was used to monitor subunit alterations in SPI after the industrial modification treatments. As shown in Figure 2, the N-SPI sample showed a normal SPI shape, with 11S and 7S subunits. The subunit composition of the 11S and 7S subunits did not change significantly after industrial modification treatments. Previous studies [29,30,31,32] showed that heat, alkali, glycosylation, and oxidative treatments did not alter the overall composition of the 11S and 7S subunits. This suggests that the four industrial modification processes do not cause changes in the subunit composition of soy globulins. Compared to N-SPI, the β-subunit band of 7S subunits in lane two becomes lighter, the 11S subunit band becomes darker, and the small molecular band of 5–17 kDa disappears. The temperatures for the denaturation of the 7S and 11S subunits were about 74 and 88 °C, respectively [18]. When the industrial heat modification treatment increased the flash temperature from 72 to 81 °C, it caused fractional disconnection of the 7S β-subunit protein assembly, but there was no severe denaturation of 11S. Thus, industrial heat modification leads to dissociation and reduced content of the 7S β-subunit in H-SPI, resulting in darker bands for the 11S subunit than for the 7S β-subunit. The heat effect of industrial heat modification promotes the formation of larger molecular weight and small molecule proteins through thiol/disulfide bond exchange reactions. Guo et al. [33] also showed that when the temperature exceeded 80 °C, large molecular aggregates were formed between SPI molecules through disulfide bonds, resulting in the disappearance of small molecular protein components. Compared to N-SPI, the A-SPI, G-SPI, and O-SPI compounds exhibited more bands in the 5–17 kDa range and above the 75 kDa range. It indicates that industrial alkali treatment, glycosylation treatment, and oxidation treatment can induce aggregation and molecular cleavage reactions in small SPI. Industrial alkali treatment facilitates the unfolding of SPI particles and the thiol oxidation retort/SH-SS interchange reaction, which forms large molecular weight protein molecules and leads to the appearance of bands above 75 kDa [34]. Alkali treatment also promotes electrostatic revulsion and subunit detachment of protein subunits to form small protein molecules, which may be responsible for the increase and disappearance of 5–17 kDa small molecule bands [31]. Industrial glycosylation treatment can promote cross-linking reactions between soybean globulin and konjac starch via Millard conjugation, forming large molecule protein bands. However, this treatment also leads to the hydrolysis of SPI, producing small molecules [30]. This result agrees with that of Diftis and Kiosseoglou [35]. In an industrial oxidative modification, free radicals can attack the 7S and 11S subunits, causing them to undergo aggregation. This is also accompanied by an attack on the peptide chains, causing them to break and forming large and small molecular weight protein bands. Wang et al. [36] also found that the oxidative attack produced protein aggregates in SPI and peptide chain breaks.

### 3.2. Influence of Industrial Adjustment Treatment on SPI Raman Spectroscopy

Figure 3 shows the Raman spectra of the SPI after the industrial modification treatments. The main band assignments include the disulfide bond region (500–550 cm^−1^), the tyrosine double chain (I_850_/I_830_), and the amide I band (1600–1700 cm^−1^). Spectral data were evaluated in 3 wavelength zones (500–550, 760–1000, and 1600–1700 cm^−1^) to study protein structure changes during processing, focusing on disulfide bond conformation, tyrosine environment, and secondary structural changes.

#### 3.2.1. Impact of Industrial Adjustment Treatment on the Secondary Structure of SPI

Changes in the amide I band between 1600 and 1700 cm^−1^ indicate alterations in protein secondary structure. The characteristic peaks at 1652, 1663, 1674, and 1683 cm^−1^ correspond to α-helix, random coil, β-sheet, and β-turn, respectively [37]. We fitted the spectra with Gaussian curves and analyzed the protein structures using Fourier transform convolution (FSD). Figure 4 shows that, compared to the N-SPI, all four industrial processing treatments decreased the quantity of α-helix and random coil structures while increasing the amount of β-sheet and β-turn structures. Notably, H-SPI showed the highest increase in β-sheet structure, while G-SPI showed the greatest increase in the β-turn form. Damodaran [38] found that β-sheet structures are more stable than α-helices and disordered structures. Previous studies have shown that α-helices and β-sheets represent regular spatial arrangements in protein structures.

In contrast, β-turns and random coils represent semi-regular arrangements [39]. Industrial heat treatment causes significant fluctuations in SPI’s polar, carbonyl, and amide groups. The hydrogen bonds that sustain the α-helix form between a carbonyl oxygen (C=O) and an amino hydrogen (NH–) break, causing protein unfolding and rearrangement. As a result, α-helical and random coil structures transform into β-folded and β-turned structures. Sun et al. demonstrated that heat treatment results in the partial folding of protein chains over a short period, increasing the quantity of β-folded structures and decreasing irregular coil forms [37]. Combined with the SDS-PAGE results, the highest increase in β-folded content may be attributed to the dissociation of the 7S subunit, which recombines and rearranges structures to form more stable intermediates. In addition, industrial alkali treatment increases the ionic strength. This enhances electrostatic revulsion among the protein molecules, which leads to protein peptide chain unfolding and α-helical structure transformation into β-turned and β-folded structures.

Li et al. reported that greater electrostatic revulsion among protein fragments might be responsible for reducing the content of irregular coils [40]. Industrial glycosylation treatment introduces konjac starch, inducing the creation of new hydrogen bonds between polysaccharides and proteins, weakening the original protein’s interaction with protein molecules, and breaking hydrogen bonds that maintain the α-helix structure. This results in the decline of α-helix quantity and the growth of β-turn and β-fold structures. Konjac starch polysaccharides induce the unfolding and denaturation of soybean-isolated proteins. The introduction of ordered konjac starch might account for the most significant increase in the β-turn structure of G-SPI [41]. Industrial oxidative modification treatment might cause the oxidation of side chain amino acids and critical peptide chains in SPI. Reactive hydroxyl radicals can modify the aromatic amino acid residues of natural proteins by forming carbon radicals [42], which can convert stable α-helix and β-fold structures into unstable ones—industrial oxidative modification treatment results in unfolding and in a more disordered structure of SPI. Cao’s study showed that free radicals generated by oxidants could attack protein molecules, inducing some degree of intermolecular aggregation [43], lessening the α-helix quantity, and expanding β-folding. The results suggest that industrial heat treatment and glycosylation treatment transform the regular helical structure of H-SPI and G-SPI protein molecules into a more extended structure, enhancing their ductility.

#### 3.2.2. Disulfide Bond Conformational Analysis

The impact of various industrial alteration treatments on the S-S conformational changes of SPI was assessed by displaying S-S stretching vibrations in the 500–550 cm^−1^ zone. The creation of different disulfide bond conformations is implicated in the S-S conformational changes. Disulfide bonds can affect protein aggregation and gel properties, as they play a potential role in protein aggregation dissemination and gel creation [44]. The majority of cysteines in the oxidized state either form intrachain disulfide bonds (S-S) inside proteins or interchain disulfide bonds among proteins, and these disulfide bonds differ depending on the conformation of the C-S-S-C atom, gauche-gauche-gauche-S-S (g-g-g, lowest molecular possible and highest constancy, 510 cm^−1^), gauche-gauche-trans-S-S (g-g-t, between uppermost and lowermost steadiness, 525 cm^−1^) and ‘trans-gauche-trans-S-S (t-g-t, stability, 545 cm^−1^). T-g-t denotes inter-chain disulfide connections among proteins, whereas g-g-g and g-g-t modes reflect intrachain disulfide bonds (S-S) inside proteins [45]. The disheveled polypeptide chain substitution observed in aberrant protein folding and subunit aggregation can be used to explain the higher quantity of t -g-g and t -g-t conformations, which imply more unstable S-S bond conformation formation [46]. A possible hypothesis is that increased protein molecule motion and collision resulting from industrial heat modification treatment can cause protein denaturation, intramolecular S-S bond cleavage, and disordered S-S bond formation. The cleavage and disordering of S-S bonds suggest a disruption in the native protein conformation, which may lead to the transformation of intra-molecular disulfide bonds (g-g-g and g-g-t) into intermolecular disulfide bonds (t-g-t). This transformation could be responsible for the observed decrease in the relative quantity of g-g-g conformation and g-g-t conformity and the increase in the relative amount of t-g-t conformity in H-SPI compared to the control group. [37]. As shown in Table 1**,** compared to the control, the g-g-g configuration of A-SPI was not significantly adjusted (*p* > 0.05), the relative quantity of the g-g-t conformation declined, and the comparative quantity of the t-g-t conformation was heightened. This may be because the disulfide bonds among the protein molecules (g-g-t) break easily in alkaline environments [47] and because of the constant generation of intermolecular disulfide bonds (t-g-t). Industrial alkali treatment gradually unfolds the protein, thus promoting the formation of S-S (g-g-t, t-g-t) bonds through thiol–thiol oxidation reactions among free thiol groups. Compared to the control, the relative contents of the g-g-g conformation and t-g-t conformation of G-SPI increased, and the g-g-t content decreased. The observed differences in the protein composition and disulfide bond formation in the konjac starch and SPI mixture may be due to the creation of new hydrogen bonds, which could cause protein dissociation or refolding. This may lead to the transformation of intramolecular disulfide bonds into intermolecular disulfide bonds [48]. Additionally, the growth in the relative quantity of the g-g-g form of G-SPI could result from covalent binding between some of the SPI and polysaccharide molecules, forming intramolecular disulfide bonds [49]. In the case of O-SPI, the relative quantity of the g-g-g conformation and t-g-t conformation was increased, and the g-g-t quantity declined. This shift in conformational content may be attributed to the protein’s α-helical structure being altered by the industrial oxidation treatment. During the treatment, the intramolecular S-S bond is broken, which results in the release of free active hydrophobic groups. These groups can then polymerize with other protein molecules, converting intermolecular disulfide bonds to intramolecular disulfide bonds and partial protein refolding [50]. Compared to the other three industrial modifications, the increase in the relative content of intermolecular disulfide bonds was smaller in O-SPI. This could be due to the industrial oxidation treatment causing intermolecular aggregation, where the sulfhydryl groups of cysteine are converted to disulfide bonds through oxygen radical attack [43].

#### 3.2.3. Amino Acid Side Chain Analysis

The ratio of the bimodal tyrosine band between 850 and 830 cm^−1^ (I_850/830_) displays polarity and hydrogen bond formation in the microenvironment surrounding the phenolic hydroxyl groups. A decline in the I_850/830_ proportion reflects a rise in the burial of Tyr residues, and an increase in the I_850_/I_830_ ratio reflects a rise in the disclosure of Tyr remains and the forming of hydrogen bonds [51]. The I_850_/I_830_ results for different industrially modified SPI samples are shown in Table 2. Compared to N-SPI, the H-SPI showed no significant change in the I_850_/I_830_ ratio (*p* > 0.05), probably because industrial heat treatment enhanced the tendency of Tyr residues to act as hydrogen bond donors to augment internal hydrogen bonding, but did not significantly alter the polarity of the tyrosine residue microenvironment [52]. Industrial alkali treatment enhanced electrostatic revulsion among SPI molecules by altering the charge environment of the protein, inducing Tyr exposure, and increasing the interchain hydrogen bonding content [53]. Li et al. [40] showed that the β-folded structure is usually stabilized by interchain hydrogen bonding. This is also consistent with the results of the transition of A-SPI from α-helix to β-fold (Figure 4). Li et al. also stated that glycosylation could lead to reduced ionization of phenolic OH oxygen, resulting in reduced hydrogen bonding between Tyr and adversely charged acceptors, such as the carboxylate ion of aspartate (Asp) or glutamate (Glu) [47]. This may be the reason for creating intermolecular hydrogen bonds between SPI and konjac starch while the exposure of Tyr residues to water continues to increase. Industrial oxidation treatment induces O-SPI structure unfolding and more Tyr exposure within the O-SPI molecule [47]. The increased exposure of tyrosine residues, which are prone to creating hydrogen bonds with H_2_O, increases the I_850_/I_830_ ratio.

### 3.3. Effects of Industrial Modification Treatment on Surface Hydrophobicity of SPI

Surface hydrophobicity characterizes the degree of exposure of internal hydrophobic groups and gelation characteristics. Hydrophobicity is the main force during gel formation [54], affecting the driving force of aggregation in the gelation process. The impacts of four industrially modified SPI on the Ho of SPI are portrayed in Figure 5. After industrial heat treatment, the protein form was disturbed and unfolded, and the protein molecules underwent conformational alterations in the secondary, tertiary, and quaternary structures [55]. This led to coverage of more hydrophobic groupings, increasing Ho. The increase in Ho after industrial alkali treatment may be due to the higher isoelectric point of the protein, resulting in greater electrostatic revulsion within the protein molecule, a looser and unfolded structure, and higher Ho. SPI typically contains 1.75–2% phytic acid, which dissociates during alkaline treatment. The acid dissociation process expands the tertiary structure of A-SPI, exposing hydrophobic residues within the molecule and improving the Ho of the sample [55]. This may account for Ho achieving the highest levels in A-SPI. Industrial glycosylation treatment may lead to the structural unfolding of soybean isolate proteins. It may expose more hydrophobic remains from the center of the molecule to the molecular exterior, which then increases the Ho value. This outcome was also verified by Raman spectroscopy analysis (Figure 4). However, adding konjac starch introduced SPI containing more hydrophilic hydroxyl groups, leading to a relative weakening of its surface hydrophobic environment [56]. This may be why the Ho of G-SPI is smaller than that of H-SPI and A-SPI. The reduction of Ho after industrial oxidation treatment may be due to the conformational fluctuations in the secondary and tertiary structures caused by reactive hydroxyl radicals. These alter the aromatic amino acid residues within the natural protein by forming carbon radicals and are continuously involved in downstream chain propagation. In addition, oxidation causes protein de-folding disorder, weak hydrogen/ion interactions, and hydrophobic interactions between surface hydrophobic groups, leading to random protein aggregation [43]. This is consistent with the lower tryptophan residue exposure and SDS-PAGE results (Figure 2). Furthermore, Baek [50] reported that after more than 40 min of oxidation, the exposed surface hydrophobic groups generated aggregates through hydrophobic interactions.

### 3.4. Effects of Industrial Modification Treatment on the Quantity of Thiol and Disulfide Bonds

The creation of aggregates in proteins is primarily attributed to disulfide bonds. Sulfhydryl groups (-SH) can affect disulfide bond formation through SH-SS exchange reactions, regulating protein aggregation and gelation [44]. Total sulfhydryl content in proteins consists of active and hidden –SH groups [18]. The SH-SS exchange reaction, which occurs in the presence of free thiols, can cause protein polymerization or block [57]. These two occurrences appear simultaneously, and their balance determines the protein’s aggregation and gel features under various circumstances. Examining different industrial modification effects on SPI’s disulfide bond and sulfhydryl content can help predict its gel network state and functional properties. As shown in Table 3, industrial heat treatment can cause protein unfolding, exposing internal SH groups and increasing the total sulfhydryl content in H-SPI. Hydrophobic groups buried in the protein molecule can promote the interaction of SH groups to shape disulfide bonds [34]. Active–SH groups may also be oxidized [58], resulting in a lower free sulfhydryl content. Industrial alkali treatment of SPI increases pH, promoting the unfolding of its spatial structure and increasing the number of anions in the solution. The increase in the number of anions contributes to the oxidation of sulfhydryl groups among some active –SH groups [59].

Some active –SH groups interact with cysteine-containing residues through S-S bonds, forming disulfide bonds. This process results in a decrease in the quantity of active –SH groups. However, the unfolding of the spatial structure exposes internal SH groups, leading to an increase in the quantity of active –SH groups [60]. This may explain the higher amount of sulfhydryl and disulfide bonds in A-SPI compared to N-SPI, with no significant change in free sulfhydryl. Industrial glycosylation of SPI dissociates or unfolds its structure, exposing SH groups and increasing the total and free sulfhydryl content [56]. Konjac starch may activate sulfhydryl functional groups, inducing new intermolecular disulfide bond bridges and leading to a rise in disulfide bond content. However, the conjugation may spatially block some SH groups on unfolded protein chains, burying some SH groups and reducing the free sulfhydryl content [61]. Industrial oxidation treatment induces the unfolding of SPI structure, and the exposed SH groups are oxidized to reversible or irreversible forms [43]. The reduction in SH and total sulfhydryl groups may be ascribed to the oxidation of the exposed SH clusters to other irreversible sulfur-containing derivatives (sulfonic and sulfinic acids) [62]. The minimal increase in disulfide bond content compared to A-SPI, G-SPI, and H-SPI may be attributed to the decrease in total sulfhydryl content. This pattern is supported by the minimal increase in the percentage of intermolecular disulfide bond conformation in Raman spectra.

### 3.5. Differential Scanning Calorimetry Analysis of SPI after Industrial Modification

The functional features of proteins could be compromised by heat treatment, resulting in protein denaturation or aggregation. In this study, differential scanning calorimetry (DSC) was employed to evaluate the degree of denaturation of different industrially modified SPI during heat treatment by measuring the energy alteration (∆H) and denaturation temperature (T_D_) of the proteins as they were continuously heated to disrupt their advanced structures. The DSC curves for the different industrially modified SPI are presented in Figure 6. N-SPI exhibited a regular DSC thermal profile, with two major exothermic transitions (peak 1 and peak 2) occurring at 72–76 °C and 91–93 °C, which were associated with denaturation at the 7S subunits (73.51 ± 0.11 °C) and the 11S subunits (91.79 ± 0.09 °C) [43]. The T_D_ values indicated the changes in thermal stability, while the changes in ∆H suggested the flexibility of the protein structure and the ease of denaturation [63]. The results in Table 4 indicate a significant reduction in 7S T_D_ and no significant change in 11S T_D_ in H-SPI compared to N-SPI.

Similarly, A-SPI exhibited no significant change in 7S T_D_ but a significant decrease in 11S T_D_. On the other hand, the T_D_ of both 7S and 11S increased in G-SPI, while the T_D_ of both the 7S and 11S subunits decreased in O-SPI. Table 4 also indicated that ∆H was significantly lower for the 7S and 11S subunits in H-SPI, A-SPI, G-SPI, and O-SPI than N-SPI. In summary, all four industrial modifications were found to alter the SPI’s thermal stability and structural flexibility, with the industrial heat modification treatment having the most significant effect on the 7S subunit structure and the industrial alkali modification treatment having the most significant impact on the 11S subunit structure. The industrial heat modification treatment caused the 7S subunit to denature at a lower temperature and become more stable and flexible when continuously heated, while 11S was less affected and remained more stable [64]. The industrial alkali treatment altered the charge distribution of SPI, disrupting its hydrogen bonds and increasing its electrostatic repulsion, leading to a more flexible structure that unfolds more easily when heated. 7S and 11S subunits were more susceptible to denaturation during continuous heating, with decreased T_D_ and ΔH [65].

The industrial glycosylation treatment process leads to the binding of SPI and konjac flour starch molecules, causing polysaccharide molecules to affect SPI molecules in space, which, to some extent, prevents the thermal aggregation of proteins. The resulting glycoprotein can maintain a more stable conformation when heated, improving the thermal stability of SPI and increasing the denaturation temperature (T_D_) of the complex. Additionally, introducing sugar molecules may break covalent or non-covalent bonds between proteins during glycosylation, causing the protein structure to become looser. As a result, the 7s and 11s subunits in G-SPI are more susceptible to unfolding during continuous heating, and the ΔH of the 7s and 11s subunits in G-SPI is reduced [66]. The industrial oxidation treatment caused SPI to unfold, with increased aggregation between protein molecules and decreased thermal stability and molecular tightness of the 7S and 11S subunits [67]. In addition, the protein was more susceptible to denaturation during continuous heating, with cross-linking between proteins and a reduction in ΔH for the 7S and 11S subunits of O-SPI. Industrial heat treatment at 81 °C caused partial denaturation of SPI, while industrial alkali, glycosylation, and oxidation treatments caused extensive denaturation. The structure of 11S was most affected by the electrostatic and hydrophobic interactions induced by the industrial alkali treatment.

### 3.6. Dynamic oscillatory Rheology

#### 3.6.1. Temperature Sweep

The rheological properties of suspensions made from SPI can be utilized to assess the impact of different industrial modification treatments on gel formation and springiness. The storage modulus G′ which measures the energy stored in a sample due to elastic deformation, provides information on the internal structure of the gel network [68]. Therefore, the elastic behavior of SPI gel is influenced by the components involved in the network and their interactions. The dynamic rheological patterns of SPI suspensions treated with various industrial modifications during heating are shown in Figure 7. Throughout the heating phase (25–95 °C), G′ begins to improve due to protein aggregation and the creation of the preliminary gel network, which may be motivated by noncovalent binding and disulfide bonds [69]. G′ continues to increase during the constant temperature (95 °C) phase as more proteins are integrated into the gel network, and network rearrangement occurs. An increase in G′ is observed during the cooling phase (95–20 °C). This gel hardening during cooling is almost thermally reversible and mainly results from increased hydrogen bond formation and Van der Waals interactions [70]. The four modifications caused the gelation temperature point of SPI G′ to decrease, implying that the SPI gelation rate is faster after undergoing these modifications. This is consistent with the DSC results, which show that the SPI samples under different industrial modification conditions have lower thermal stability and denaturation temperatures than the N-SPI samples. Gosal et al. showed a mutation point in the logarithmic level of G′ close to 90°, when the globulin starts to form a gel in response to heat, indicating that the protein transitions from liquid to solid gel [71]. The temperature of this mutation point is known as the gelation temperature point. For industrial-modification-treated SPI, the final G′ for gel formation and the initial G′ values are much higher than those of N-SPI. As the initial G′ value decreases, the gelation temperature point decreases to 86 °C. The final G′ and initial G′ values for gel formation are much higher than those of N-SPI due to the four industrial modifications, which expose a higher number of buried hydrophobic and hydrogen bonds, as well as disulfide bonds involved in intermolecular interactions during gel structure formation and consolidation [72]. This pattern is supported by the Ho (Figure 5) and disulfide bond (Table 3) data. The preliminary G value reduction is due to aggregate depolymerization, resulting in decreased springiness. A-SPI’s higher surface hydrophobicity and internal hydrophobic groups cause a more significant increase in G′ value than H-SPI. This increases non-covalent bonding interactions between molecules, leading to enhanced aggregation and better gel network creation. The voids in the gel matrix are occupied mainly by H_2_O and soluble proteins [73]. At the same time, the konjac starch in the industrial glycosylation-treated SPI system acts as a filler phase that absorbs water to swell and form gels, leading to a more ordered gel network structure that prevents water loss and creates a stronger matrix [74]. Additionally, the loss of secondary structure due to industrial glycosylation, protein structure folding, and depolymerization of macromolecules enhances noncovalent interactions among protein molecules during the three stages of gel formation, leading to the largest G′ values [75]. However, the highest initial G′ value for gel formation is observed in industrially oxidized SPI, with a decline in the initial G′ value and a decrease in the gelation temperature point to 88 °C. This is due to the effective unfolding of SPI under industrial oxidation treatment and the rise in disulfide bonds at the end of the heating stage [36]. The final G′ value is smaller than that of H-SPI, A-SPI, and G-SPI, probably because oxidation causes protein de-folding disorder and weak hydrogen/ion interactions, leading to random protein aggregation and a weakened gel structure.

#### 3.6.2. Frequency Sweep

The appearance of the gel is an essential sensory characteristic that can impact consumer acceptance. As depicted in Figure 8, the four industrially modified SPI-produced gels that were more transparent compared to the N-SPI gels. Notably, the gels made from H-SPI and G-SPI were smoother and more transparent. The color values are known to be associated with cross-link and bond formation, which results in the formation of dense structures. The ability to scatter light is stronger when the gel networks form a large and widely connected region. Conversely, the light scattering ability decreases when the gel networks form fewer crosslinks [76]. Therefore, the thermally induced gels made from H-SPI and G-SPI, with smoother surfaces and higher transparency, have stronger gel networks.

We investigated the impact of various industrial conditions on SPI gels after treatment. The graphs in Figure 9 illustrate the results of an SPI frequency sweep following the temperature sweeps. The values suggest that the G′ and G″ of H-SPI, A-SPI, and G-SPI slightly increased with frequency, while O-SPI did not change significantly. Additionally, G′ and G″ increased for all industrially modified SPI compared to N-SPI. G-SPI gels exhibited the highest G′ and G″. Notably, G′ was greater than G″ for all protein samples. According to Clark and Hodgson [77,78], sharper gels have G′ levels that are consistently superior to G″ rates across the entire frequency range. This aligns with our observations, suggesting that our samples belong to the stronger gels category. SPI formed stronger gels under industrial heat treatment compared to N-SPI. This may be attributed to the higher capacity of industrially heat-treated SPI to form protein interactions, potentially in the form of hydrophobic bindings (Figure 5) and disulfide bonds (Table 3). The increased content of A-SPI disulfide bonds from industrial alkali treatment can also explain these interactions similarly [79]. Industrial glycosylation modification showed the most significant improvement in SPI gelation, consistent with the high disulfide bond content in G-SPI.

In contrast, konjac starch polysaccharide chains can bind around SPI via hydrophobic interactions and hydrogen bonds. The increased quantity of protein aggregates with high disulfide bonding ability was more noticeable under industrial glycosylation treatment. Covalent bonds were the primary protein interactions, forming covalently bound aggregates with a more stable structure and higher G′ vs. G″ [79]. The hydrophilic groups of konjac starch polysaccharides could type an ordered 3D-network structure with proteins via hydrogen bonding, stimulated dipoles, molecular dipoles, and transient dipoles during gelation, leading to the highest G′ vs. G″ [22]. Following industrial oxidation treatment, G′ and G″ of O-SPI were less impacted by frequency, and G′ increased. This was due to the promotion of protein refolding and the cross-connecting of protein side chains by industrial oxidation treatment, enhancing hydrophobic and disulfide bond interactions and forming stronger gels [80]. The hydrophobic and disulfide bonding results also correspond to the G′ vs. G″ values of only O-SPI, which are higher than N-SPI.

### 3.7. Features of MP-SPI Composite Gel

#### 3.7.1. Profile Analysis

TPA parameters were used to evaluate the strength of MP and composite gels by mixing MP with N-SPI, H-SPI, A-SPI, G-SPI, and O-SPI. Table 5 tabulates the fluctuations in TPA factors involving hardness, springiness, and cohesiveness properties. The supplement of N-SPI reduced the hardness, springiness, and cohesiveness properties of MP gels. However, when the four industrially modified SPI were added to MP, their hardness and springiness properties were significantly higher than those of MP gels and MP-NSPI mixed gels. Still, there was no significant difference in their cohesiveness properties.

Moreover, the addition of industrial alkali-modified treated A-SPI showed the best improvement in the hardness and springiness of the combined gels. After adding N-SPI, it was dispersed through the protein gel network, which may have disrupted the protein binding in the gel substance, hindering gel network creation and resulting in weaker mixed gels. Sun and Arntfield [72] demonstrated that the spherical structure of hard packing could prevent the interaction. Industrial heat modification treatment increases protein molecular motion and collision probability and induces protein denaturation, increasing hydrophobic interactions. This promotes cross-linking between H-SPI and MP, leading to a denser gel network structure and increased hardness and springiness.

Furthermore, the secondary structure changes showed (Figure 4) that H-SPI had higher β-folding content than the control group, and β-folding is essential for creating the gel network due to its large surface area with ordered hydrogen bonds and weak hydration strength [81]. The higher β-folding content in H-SPI results in increased hydrogen bonds during gelation, leading to a denser gel network structure with cross-linking between H-SPI and MP. This leads to enhanced hardness and springiness. Industrial alkali treatment can also cause partial unfolding of SPI, exposing reactive groups, which yields equivalent results to industrial heat modification treatment. MP-ASPI has higher hardness, springiness, and chewiness than MP-HSPI due to the stronger interaction and/or reactivity of A-SPI with MP, which promotes cross-linking between the two and leads to a denser network structure. The introduction of sugar chains by industrial glycosylation treatment results in a more ordered gel network due to increased resistance to molecular motion [82]. Protein aggregates are dissociated or unfolded, exposing hydrophobic groups, which enhance cross-linking between G-SPI and MP, forming a denser gel network. Industrial oxidation treatment causes the partial unfolding of proteins and the disclosure of hydrophobic clusters [43], leading to oxidative stress. However, O-SPI easily forms poor aggregates, which weakens the interaction force between O-SPI and MP, reducing their aggregation and forming a gel network with varying pore sizes and disorganization. O-SPI also has lower thermal stability and is more prone to aggregation or denaturation [83]. As a result, the textural properties of MP-OSPI composite gels are only higher than those of single MP gels and MP-NSPI gels.

#### 3.7.2. WHC of MP Composite Gels

The water holding capacity (WHC) of different industrially modified SPI and MP composite gels was evaluated. The water retention results of the MP-SPI composite gel and the MP gel are shown in Figure 10, with MP single gel as the control. The results showed that WHC content decreased when N-SPI was mixed into MP-SPI gel. However, four different types of industrially modified SPI were added, and their WHC was significantly increased. In addition, after adding G-SPI, MP-GSPI composite gels had the highest WHC. The change in WHC is closely related to protein cross-linking, hydrophobic interaction, and hydrogen bonding [84]. Adding N-SPI reduced the WHC of MP-NSPI gels because the self-association of two proteins increased, forming their gel blocks, leading to an uneven and weak gel network with a large pore network structure and reducing the retention rate of water molecules [72]. Industrial heat modification treatment increases protein molecules’ movement rate and collision probability, inducing protein expansion and exposing more active sites or regions [85]. The interaction between hydrophobic and disulfide bonds is enhanced, which may lead to A-SPI interacting more easily with MP, forming a closer gel network and thus capturing more water. Industrial alkali treatment changes the pH environment, changes the charge distribution of the SPI amino acid side chain, induces the expansion of SPI, and increases the exposure of hydrophobic groups in SPI [86]. The strongest hydrophobic and disulfide bond interactions occur, enhancing the electrostatic revulsion between MP and SPI, causing a more uniform protein network with lesser aperture and more continuity, which captures more water and improves WHC [87]. Industrial glycosylation treatment induces the dissociation or expansion of protein aggregates, which exposes protein hydrophobic groups and enhances the hydrophobic force between protein and water. The strongest disulfide bond interaction occurs in G-SPI, promoting covalent crosslinking between MP and G-SPI, forming a well-structured gel network, intercepting water molecules, and improving WHC. Also, creating hydrogen bonds between hydrophilic polysaccharides and water molecules can improve WHC [88]. Industrial oxidation treatment induces the unfolding of SPI. Although the hydrophobic and disulfide bond bindings are weak, the increased flexibility of the SPI structure and the decreased structural compactness after industrial oxidation are conducive to crosslinking with MP, forming a more compact gel network, retaining more H_2_O molecules, and improving WHC [84].

#### 3.7.3. Rheology of MP-SPI Gel Analysis

The rheological features of the MP-SPI composite gel system were investigated by determining the shift in the storage modulus (G′) through heating. G′ represents the energy stored in the gel sample through heating and is an essential dynamic rheological indicator that reflects its elastic actions [89]. MP undergoes three stages during gel formation via heating from 20 to 80 °C, including unfolding, aggregation, and network structure formation [90]. The gel configuration of blank MP samples happened at 40 °C, as shown in Figure 11, where G′ began to increase, reaching its highest point at 49 °C. The rise in G′ from 38 to 49 °C is described as a “gel setting” for forming a loose gel structure. G′ then dropped sharply to a trough at 54 °C, and the decrease in G′ from 49 to 54 °C is referred to as “gel weakening.” Subsequently, during heating at 54 °C, G′ again steadily increased. The helix-to-helix transition of myosin disrupted the formerly formed protein network [91]. As shown in Figure 11, the G′ of MP-NSPI was lower than that of MP at temperatures varying from 20 to 80 °C, while the G′ of MP-HSPI, MP-GSPI, and MP-OSPI was higher than that of MP at temperatures varying from 20 to 80 °C. In addition, the G′ of MP-ASPI was initially lower than that of MP until 20–54 °C but then increased rapidly after 54 °C and had the highest G′ among all samples after 68 °C.

A final temperature of 65–74 °C was satisfactory for general meat processing circumstances. Nevertheless, this temperature was inadequate to denature 100% of the N-SPI protein structure. This may clarify why N-SPI did not contribute to the gel properties of MP-NSPI. N-SPI has a tightly packed spherical structure that prevents interaction with meat proteins, resulting in reduced cross-links among protein aggregates [19]. This may be why the G′ of MP-NSPI is lower than MP at both 20 and 80 °C. The hydrophobic group is the main site of binding between the two proteins, MP and SPI, and the industrial heat treatment unfolded the SPI, exposing the buried hydrophobic group. As a result, the hydrophobic interaction between SPI and the MP heavy chain is enhanced, and a higher G′ appears. This behavior is supported by the hydrophobic (Figure 5) results. Industrial alkali treatment induced the exposure of the reactive side chain groups present in the SPI and increased with the growth in temperature during heating into the molten sphere state. In the molten globule state, A-SPI exposed more hydrophilic amino acid residues and formed more protein aggregates. A-SPI aggregates also had a repulsive volume effect, accelerating the development of a denser and more ordered gel network [92]. Industrial glycosylation treatment increased the number of hydrophobic groups in the long chains of SPI, resulting in enhanced hydrophobic interactions. At the same time, konjac starch dissolves in water to form gels easily, and the gels formed after cross-linking with MP will have higher springiness [93]. In addition, the highest disulfide bond content results showed that the enhanced disulfide bonding by G-SPI may be the reason for the increased gel ability. In addition, the more stable structure of sugar molecules on protein chains and the inherent spatial site resistance could prevent intermolecular aggregation and new protein bond formation, which could explain the lower G′ values when compared to MP-ASPI after 68 °C [94]. This pattern is also supported by the results of Ho. More hydrophobic groups and lower thermal stability of A-SPI than G-SPI, stronger hydrophobic interaction, and easier thermal denaturation result in MP-ASPI having a higher G′ value than MP-GSPI; it has the highest G′ of all samples after 68 °C. Industrial oxidation treatment can oxidize key peptide chains and side chain moieties of SPI, thus supporting the development of the protein structure and leading to the exposure of hydrophobic sites [95]. This may be the reason for the lower G′ of MP-OSPI compared to MP and MP-NSPI. In contrast, the SDS-PAGE and Raman spectroscopy results of the secondary structure of O-SPI indicated that O-SPI was more inclined to self-conjugation, which might be why the lower G′ of MP-OSPI as compared to MP-HSPI, MP-GSPI, and MP-ASPI.

#### 3.7.4. SEM

The functional features of gels, such as WHC, strength, and rheology, are determined by their three-dimensional networks. The physical features of the gel structures are closely related to their microstructures. The structures of composite gels made from SPI and MP after various industrial modifications were observed using scanning electron microscopy (SEM) (×1000) (Figure 12). Associated with the MP gels, the roughness of the MP-NSPI gel network increased, and the internal pores became larger and not uniformly distributed. However, all four industrial modifications followed in a denser MP-SPI composite gel structure and a more well-defined interfacial network. After comparison, it can be found that the industrial alkali modification treatment has the best improvement effect on the composite gel, and the composite gel structure formed is the most uniform and dense. There is irregular collapse on the surface of the composite gel produced by MP and N-SPI, and the cut surface contains many lamellar structures, spherical aggregates, and an obvious sense of fault. This may be due to the circular structure of N-SPI hard packing occupying the MP intermolecular binding sites and weakening its network structure [4]. The development of the gel microstructure mainly depends on the relative speeds of protein unfolding and aggregation. When the aggregation speed is faster than the unfolding speed, it forms a dense and uniform gel microstructure. When the aggregation rate is lower than the unfolding rate, it forms rough and diverse gel microstructures [96]. Considering unfolded proteins, N-SPI molecules cause MP molecules to aggregate much slower, and the aggregation speed is lower than the unfolding speed. Therefore, protein fragments aggregate to form spherical aggregates and irregular networks. The irregular cavities on the surface of MP-HSPI composite gels were significantly reduced, and most were dense network structures. The irregular cavities on the surface of MP-HSPI composite gels may be attributed to the temperature below 90 ℃. The 11S subunit was unexpanded, and only the 7S subunits unfolded; there was the incomplete binding of MP to H-SPI, and there was the formation of a partially rough and diverse gel network structure. MP-ASPI gels have fewer irregular pores on the surface, dense structures, and very well-defined gel interface networks. Combined with the analysis of the DSC results (Table 4), A-SPI has the lowest overall enthalpy change at the 7s and 11s subunits. Therefore, it requires the lowest energy for unfolding and can unfold faster to form aggregates with MP, and composite protein fragments aggregate to form flattened aggregates and regular networks.

In addition, more hydrophobic residues (highest Ho, Figure 5) and sulfhydryl groupings are visible, and protein–protein interactions are improved, contributing to the formation of a homogeneous network [27]. The surface of MP-GSPI gels is more regular, with a high level of protein cross-linking and the appearance of striated structures on the surface. This can also be attributed to the decrease in the content of G-SPI active SH groups and the exposure of hydrophobic parties on the protein surface (Figure 5, Table 3). This promotes intermolecular interactions to form tighter networks, and more active sulfhydryl groups involved in creating disulfide bonds through hydrophobic binding form tighter gel structures. The appearance of the striped structure may be attributed to the introduction of sugar chains [97]. MP-OSPI gels have a more regular surface, smaller internal pores, and irregular structures. This may be ascribed to the aggregation among O-SPI molecules, forming partial O-SPI aggregates and affecting the MP aggregation rate, which is lower than the unfolding rate, forming rough and diverse gel microstructures [98]. In addition, it may cause the formation of spherical aggregates and irregular networks.

## 4. Conclusions

This study examined the impacts of four industrial modifications on the gel properties of SPI and the structure of SPI, as well as its complexation with MP in the preparation of protein gels. None of the industrial modifications significantly influenced the subunit composition of SPI. However, they increased the content of β-fold and β-turned structures and decreased thermal stability. All of the modifications decreased the gelling temperature point and G′ of SPI, resulting in faster protein aggregation. The gel intensity, WHC, and texture features of the MP-SPI composite gels were significantly enhanced by the modifications, particularly the alkali modification treatment. The addition of N-SPI disrupted myosin interactions and adversely affected the formation of MP gels. The four industrial modification treatments can be applied to SPI to improve its gelation ability. We observed that MP-ASPI had the highest gel strength and the most continuous, dense gel network structure. Therefore, a-SPI could be used as a potential functionality-enhanced soy protein ingredient in comminuted meat products. The research results will guide food companies to develop industrial modification processes based on laboratory test results, provide high-quality raw material production methods for producing high-value SPI products, and have important practical significance for efficiently using the variety of soybean resources in the future market economy.

## Figures and Tables

**Figure 1 foods-12-01982-f001:**
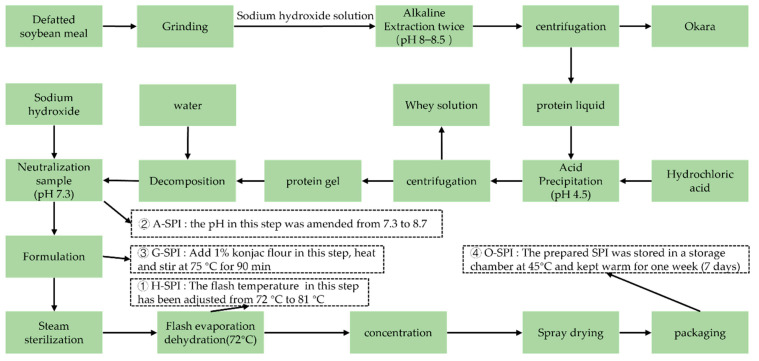
The industrial production process of SPI.

**Figure 2 foods-12-01982-f002:**
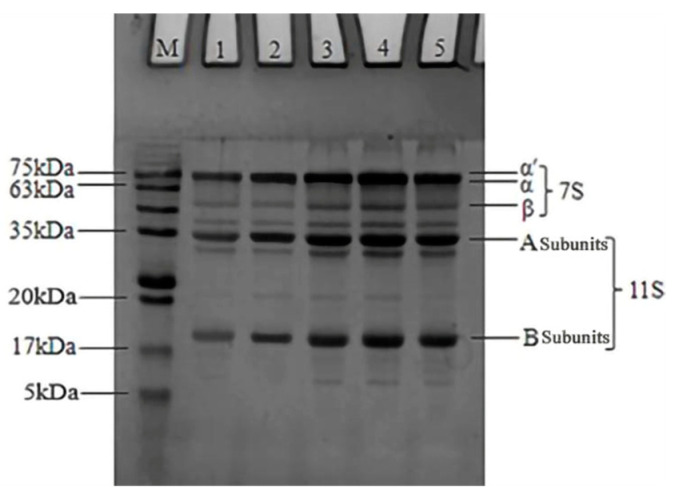
SDS-PAGE results of SPI after different industrial modification treatments. Note: M: standard protein marker, 1: N-SPI, 2: H-SPI, 3: A-SPI, 4: G-SPI, 5: O-SPI.

**Figure 3 foods-12-01982-f003:**
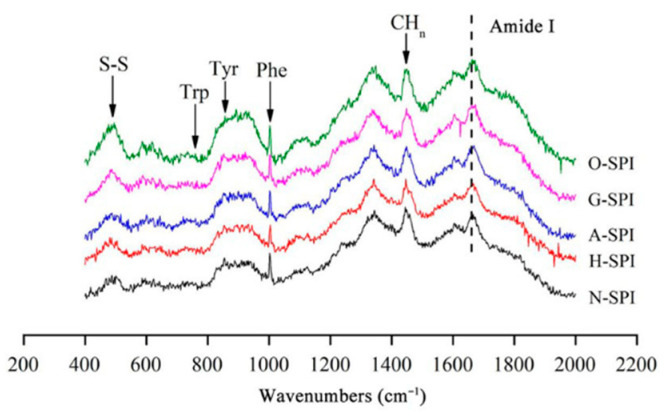
Raman spectra of SPI after different industrial modification treatments.

**Figure 4 foods-12-01982-f004:**
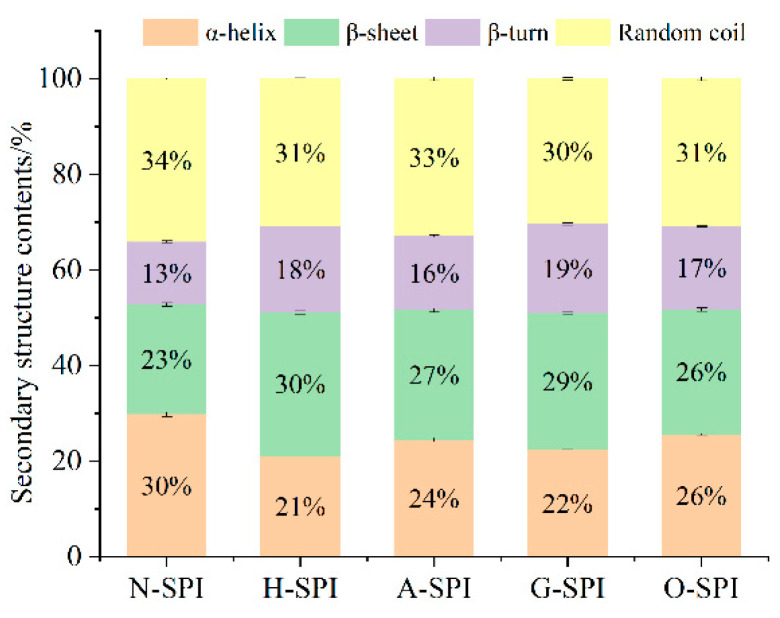
Fitting results of the secondary structure of different industrial modified SPI by amide I band.

**Figure 5 foods-12-01982-f005:**
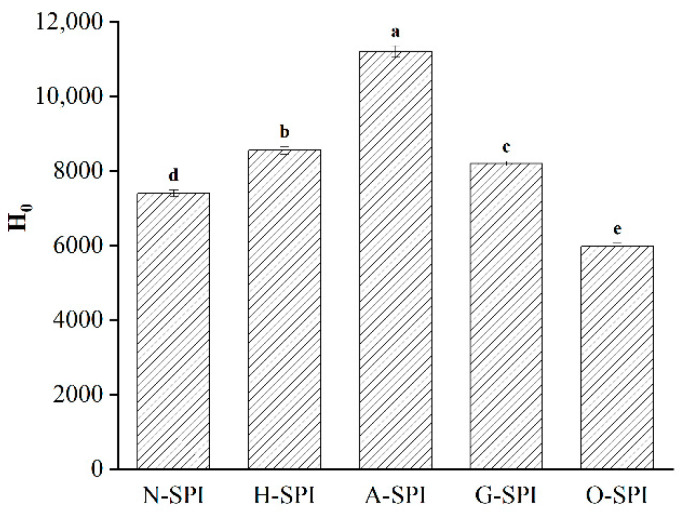
Effect of different industrial modifications on surface hydrophobicity of SPI. Note: Different letters (a~e) in the same column indicate significant differences (*p* < 0.05).

**Figure 6 foods-12-01982-f006:**
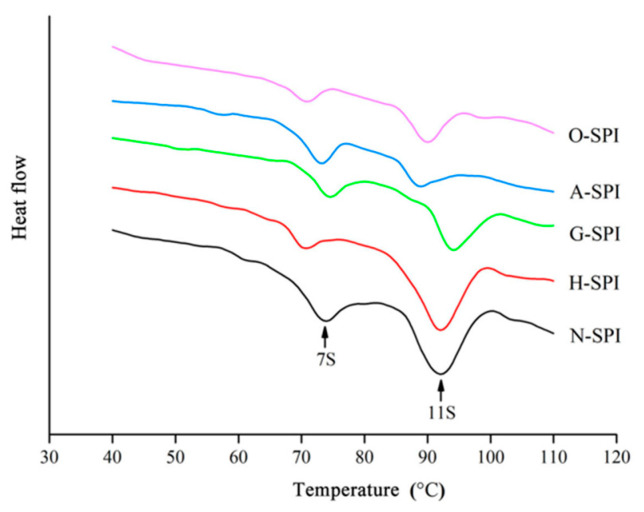
DSC thermograms of different industrially modified SPI.

**Figure 7 foods-12-01982-f007:**
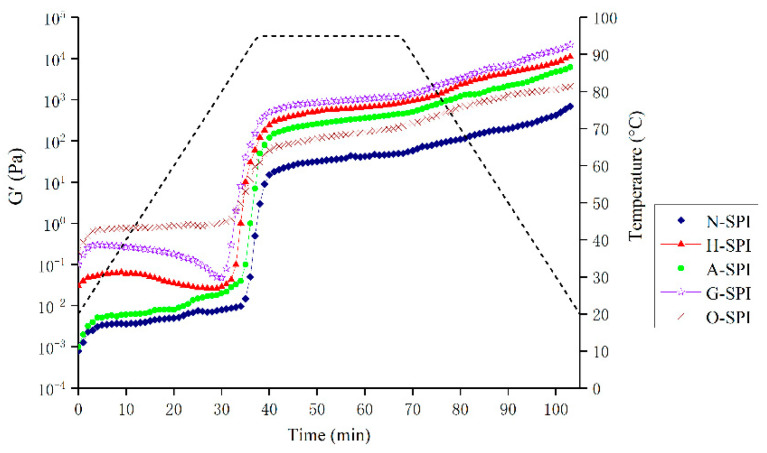
G′ as a function of heating time and temperature in the process of SPI gel formation.

**Figure 8 foods-12-01982-f008:**
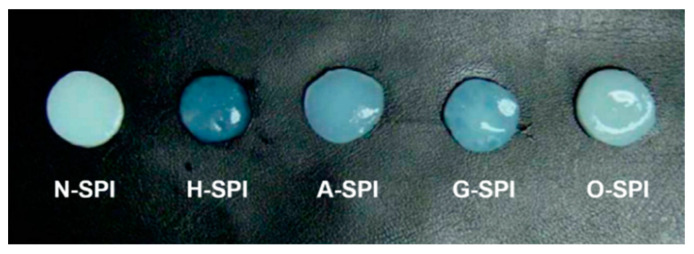
Pictures of protein gels formed by different industrially modified SPI.

**Figure 9 foods-12-01982-f009:**
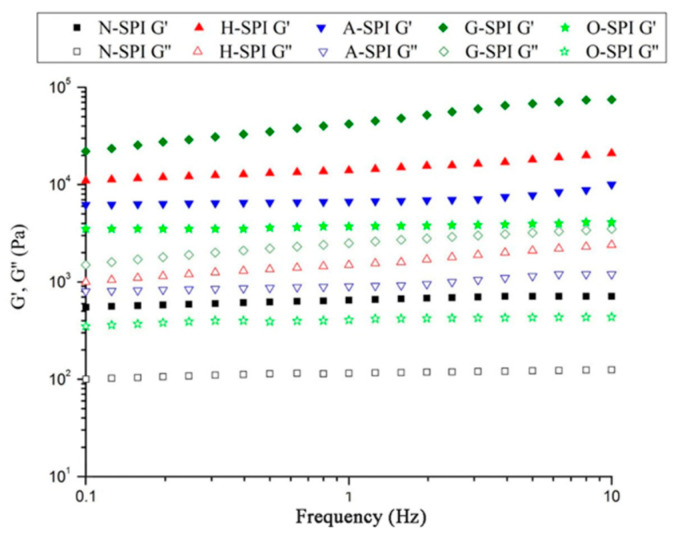
G′ and G″ as a function of frequency for SPI after temperature sweep.

**Figure 10 foods-12-01982-f010:**
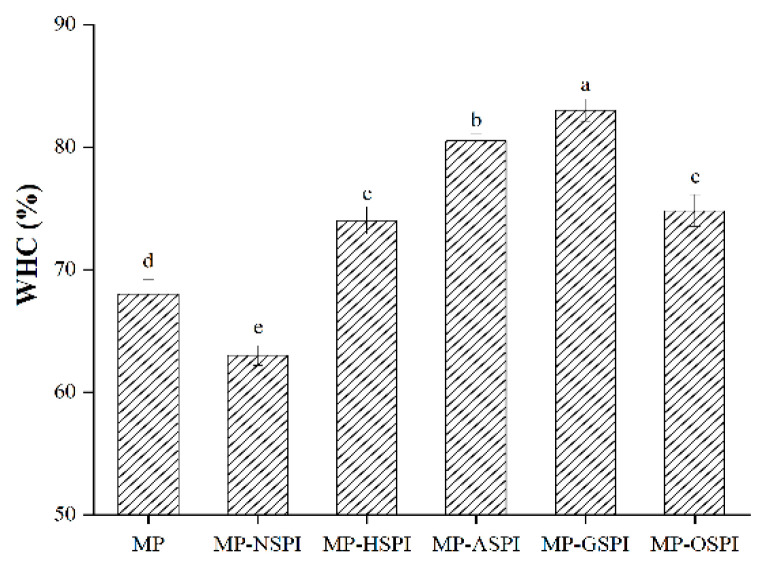
The WHC of different industrially modified SPI and MP composite gels. Note: Different letters (a~e) in the same column indicate significant differences (p < 0.05).

**Figure 11 foods-12-01982-f011:**
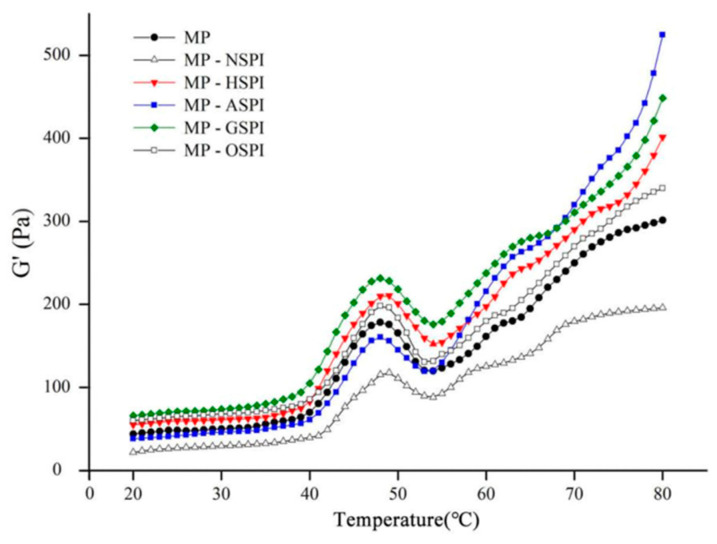
Storage moduli (G′) as a function of temperature in the process of MP-SPI composite gel formation (MP: SPI = 4:1, total protein 3.75%).

**Figure 12 foods-12-01982-f012:**
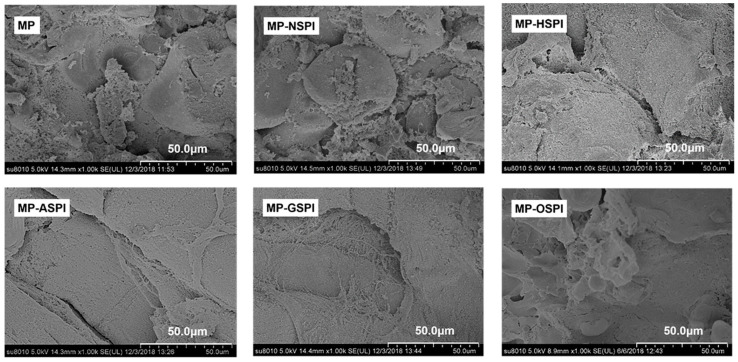
The scanning electron microscope results of MP gel and MP-SPI composite gel (SEM images with 1000× of magnification).

**Table 1 foods-12-01982-t001:** Effects of different industrial modifications on the conformation of the S-S bonds of SPI.

Samples	Disulfide Bond Configuration		
	g-g-g (%)	g-g-t (%)	t-g-t (%)
N-SPI	44.32 ± 0.48 ^c^	38.50 ± 0.16 ^a^	17.18 ± 0.29 ^c^
H-SPI	39.82 ± 0.43 ^d^	36.12 ± 0.39 ^b^	24.06 ± 0.32 ^a^
A-SPI	44.75 ± 0.42 ^c^	31.44 ± 0.41 ^d^	23.81 ± 0.22 ^a^
G-SPI	46.41 ± 0.21 ^a^	30.29 ± 0.34 ^d^	23.30 ± 0.13 ^a^
O-SPI	45.99 ± 0.17 ^b^	34.72 ± 0.34 ^c^	19.29 ± 0.20 ^b^

Note: Different letters (a~d) on the same column indicate significant differences (*p* < 0.05).

**Table 2 foods-12-01982-t002:** Analysis results for the I_850_/I_830_ intensity ratio of different industrially modified SPI.

Sample	Tyrosine Fermi Resonance line I_850_/I_830_
N-SPI	1.007 ± 0.002 ^c^
H-SPI	1.006 ± 0.001 ^c^
A-SPI	1.015 ± 0.001 ^b^
G-SPI	1.025 ± 0.001 ^a^
O-SPI	1.016 ± 0.002 ^b^

Note: Different letters (a~c) in the same column indicate significant differences (*p* < 0.05).

**Table 3 foods-12-01982-t003:** Effect of different industrial modifications on SPI disulfide bond and sulfhydryl content.

Sample	Disulfide Bond Content (μmol/g)	Total Sulfhydryl (μmol/g)	Free Sulfhydryl (μmol/g)
N-SPI	1.00 ± 0.01 ^e^	3.64 ± 0.01 ^b^	1.63 ± 0.02 ^a^
H-SPI	1.38 ± 0.01 ^b^	4.15 ± 0.02 ^a^	1.38 ± 0.02 ^b^
A-SPI	1.28 ± 0.01 ^c^	4.20 ± 0.03 ^a^	1.64 ± 0.01 ^a^
G-SPI	1.40 ± 0.02 ^a^	4.21 ± 0.03 ^a^	1.41 ± 0.03 ^b^
O-SPI	1.13 ± 0.01 ^d^	3.24 ± 0.01 ^c^	0.98 ± 0.01 ^c^

Note: Different letters (a~e) in the same column indicate significant differences (*p* < 0.05).

**Table 4 foods-12-01982-t004:** Denaturation temperature (T_D_) and enthalpy (∆H) of different industrially modified SPI.

Sample	DSC Parameter			
	T_D_ 7S (°C)	T_D_ 11S (°C)	ΔH 7S (J/g)	ΔH 11S(J/g)
N-SPI	73.51 ± 0.11 ^b^	91.79 ± 0.09 ^b^	1.63 ± 0.01 ^a^	7.98 ± 0.01 ^a^
H-SPI	70.36 ± 0.05 ^d^	91.52 ± 0.18 ^b^	1.03 ± 0.02 ^d^	6.25 ± 0.05 ^b^
G-SPI	74.69 ± 0.02 ^a^	93.20 ± 0.03 ^a^	1.43 ± 0.05 ^b^	5.34 ± 0.03 ^c^
A-SPI	73.22 ± 0.05 ^b^	88.18 ± 0.15 ^d^	1.47 ± 0.01 ^b^	3.41 ± 0.12 ^d^
O-SPI	71.72 ± 0.06 ^c^	90.31 ± 0.10 ^c^	1.22 ± 0.02 ^c^	5.21 ± 0.08 ^c^

Note: Different letters (a~d) in the same column indicate significant differences (*p* < 0.05).

**Table 5 foods-12-01982-t005:** The results of MP-SPI composite gel texture characteristics.

Sample	Texture Properties		
	Hardness (N)	Springiness (mm)	Cohesiveness (mJ)
MP	277.3 ± 2.2 ^e^	0.88 ± 0.01 ^d^	0.54 ± 0.02 ^b^
MP-NSPI	253.4 ± 6.2 ^f^	0.83 ± 0.02 ^e^	0.58 ± 0.01 ^a^
MP-HSPI	316.3 ± 3.0 ^c^	0.91 ± 0.01 ^bc^	0.59 ± 0.01 ^a^
MP-ASPI	394.7 ± 5.5 ^a^	0.96 ± 0.02 ^a^	0.59 ± 0.01 ^a^
MP-GSPI	344.0 ± 3.6 ^b^	0.92 ± 0.01 ^b^	0.60 ± 0.02 ^a^
MP-OSPI	292.5 ± 4.5 ^d^	0.90 ± 0.01 ^c^	0.60 ± 0.02 ^a^

Note: Different letters (a~f) in the same column indicate significant differences (*p* < 0.05).

## Data Availability

The data presented in this study are available on request from the corresponding author.

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
