# Peer review of "Impacts of Industrial Modification on the Structure and Gel Features of Soy Protein Isolate and its Composite Gel with Myofibrillar Protein"

_foods, 2023, doi:10.3390/foods12101982_

Round 1
Reviewer 1 Report
The Article (article) is very interesting and relevant for readers of the journal "FOODS". It addresses the formulation of different systems derived from soy proteins and their mixture with myofibrillar proteins. Soy proteins were modified by different physical-chemical methods (heating, alkaline solubilization, glycolization, oxidation). Different experiments were carried out that provided a very broad set of data on these systems and that greatly contributed to the development of food ingredients with potential industrial application.
However, the wording is confusing in some parts, in others some paragraphs seem to make no sense and in different points technical terms are referenced in an unusual way which suggests the need for a thorough revision of the English edition.
In this way, I make some specific suggestions on more evident issues.
Abstract:
- Line 13: sentence needs revision.
- Ideas seem out of order, which makes it difficult to understand: suggestion to first cite the effect of the process and define the letters H, A, G and O as abbreviations for the type of treatment (definition was not presented at the beginning of the abstract). Then describe the effect on the soy protein fractions and then present the effects on the structure of the mixed gels.
- lines 28 to 30 - Review ideas and editing in English. The sentence seems to make no sense like this.
- Line 32 “low-gelling properties of soy protein” I believe you mean native soy proteins.
- Line 38 “solidify matrix” – solidified matrix?
- Line 52: durable state or gel- I believe you mean hard gels or self-sustained gels
- Line 70: What is “flexible modulus”? Do you mean elastic modulus?
- Lines 79-96: At the end of the introduction, a long section is devoted to the specific description of the present work approach and a predictive description of the obtained results, however, the suggested modifications for soy proteins were not detailed. I believe the whole section in lines 79 to 95 could be more adequately arranged in the discussion section and the author shod concentrate at this point of the text to provide more references to support the treats applied to change the structure and therefore protein functionality.
- - Line 105- Figure 1 and treatments: Figure 1 and relationship with lines 109 to 111: Correlating the modification processes with the figure is difficult. I understand that the stages where the changes occur have been demarcated, however, it is unclear. For each process the author quotes “while other parameters remained unchanged”, but what are the true altering routes for each method? One suggestion would be to change the figure so that the production routes would be more demarcated. Arrows or grids in specific colors could achieve this by demarking different routes. It is very confusing and difficult to understand the reproducibility of treatments. Additionally, provide the most specific details of each treatment method so the reader can compare your data with those of other authors, reproduce experiments more assertively and build new hypotheses. This is the main point of your work and details are needed.
- line 109: Reference 13 cited in this paragraph is somewhat general. One approach is to utilize a separate reference for each technique that lists the exact treatments that were performed. The observation that these modifications are portrayed as regular industry operations, which does not seem as obvious, nonetheless raises this point.
- Laemeli's approach is quite thorough and includes specifics for each desired interaction in the aggregates (SDS-PAGE lines 124 to 126). Even though it is a well-established and widely used procedure, the author still needs to provide these facts.
- Lines 202-206- TPA method: Describe the parameters of compression test experiments in TPA mode. This method involves applying very specific conditions that were not defined by the author.
- Item 2.13 could be incorporated into item 2.8, noting the description of the different rheological parameters used for the SPI and PM systems.
- The text needs a detailed review of the English editing, and some technical terms seems wrong. The author uses terms I'm not used to, and I was particularly intrigued by the rheological study. To name a few cases: - flexible modulus not the place of elastic or storage module? - accumulation rather than aggregation? - mutation temperature instead of transition temperature or gelation temperature? - gels are faster instead of gelation rate is faster.
- Lines 592-595- it would be interesting to revise. The idea seems to be of a result obtained but at the end of the paragraph it is understood that it refers to the result of another cited work (reference 15) to whom you intended to correlate the data
- - line 600 “.in industrially oxidized ” seems a typing error
- Line 622 – “gel sweep tests” instead of “frequency sweeps”
The article requires extensive editing, especially in terms of paragraph construction and the standardization of technical terms to what is commonly used in the literature.
Despite these points, the article is interesting and appealing to “FOOD” readers. Therefore, I recommend that the author should be advised to perform an English editing, revise, and standardize the technical terms, and reformulate the work in a new submission
Author Response
Dear Editors and Reviewers:
Thank you for your letter and for the reviewers’ comments concerning our manuscript entitled “Impacts of Industrial Modification on the Structure and Gel Features of Soybean Protein Isolate and its Composite Gel with Myofibrillar Protein” (ID: foods-2345837). Those comments are all valuable and very helpful for revising and improving our paper, as well as the important guiding significance to our researches. We have studied comments carefully and have made correction which we hope meet with approval. Revised portion are marked in red in the paper. We hope that the revision is acceptable and look forward to hearing from you soon. The main corrections in the paper and the responds to the reviewer’s comments are as following:
Reviewer #1:
The Article (article) is very interesting and relevant for readers of the journal "FOODS". It addresses the formulation of different systems derived from soy proteins and their mixture with myofibrillar proteins. Soy proteins were modified by different physical-chemical methods (heating, alkaline solubilization, glycolization, oxidation). Different experiments were carried out that provided a very broad set of data on these systems and that greatly contributed to the development of food ingredients with potential industrial application.
However, the wording is confusing in some parts, in others some paragraphs seem to make no sense and in different points technical terms are referenced in an unusual way which suggests the need for a thorough revision of the English edition.
In this way, I make some specific suggestions on more evident issues.
Abstract:
Line 13: sentence needs revision.
Response: Thanks for your valuable suggestions. We have deleted it and “The influence of different industrial modification methods (heat (H), alkaline (A), glycosylation (G), and oxidation (O)) on the structure of SPI, the properties of gel and the gel properties of soy protein isolate (SPI) in myofibril protein (MP) was evaluated.” have been added and marked in red. (Line 6-8 in new version).
Ideas seem out of order, which makes it difficult to understand: suggestion to first cite the effect of the process and define the letters H, A, G and O as abbreviations for the type of treatment (definition was not presented at the beginning of the abstract). Then describe the effect on the soy protein fractions and then present the effects on the structure of the mixed gels.
Response: Thanks for your valuable suggestions. We have restructured the abstract to first define the abbreviations used and cite the effect of the process, followed by a description of the effect on the soy protein fractions and then the presentation of the effects on the structure of the mixed gels. (Line 6-7 in new version).
We have modified it to” Native soy protein isolate (N-SPI) has a low denaturation point and low solubility, limiting its industrial application. The influence of different industrial modification methods (heat (H), alkaline (A), glycosylation (G), and oxidation (O)) on the structure of SPI, the properties of gel and the gel properties of soy protein isolate (SPI) in myofibril protein (MP) was evaluated. The study found that four industrial modifications did not influence the subunit composition of SPI. However, the four industrial modifications altered the secondary structure and the disulfide bond conformation content of SPI. A-SPI exhibits the highest surface hydrophobicity and I850/830 ratio, but the lowest thermal stability. G-SPI exhibits the highest disulfide bond content and the best gel properties. Compared with MP gel, but the addition of H-SPI, A-SPI, G-SPI, and O-SPI components significantly improved the properties of the gel. Additionally, MP-ASPI gel exhibits the best properties and the microstructure. Overall, the four industrial modification effects may impact the structure and gel properties of SPI in different ways. A-SPI could be used as a potential functionality-enhanced soy protein ingredient in comminuted meat products. The present study results will provide a theoretical basis for industrialized production of SPI.”
lines 28 to 30 - Review ideas and editing in English. The sentence seems to make no sense like this.
Response: Thanks for your valuable suggestions. I've deleted it and the necessary revisions have made
“Soy protein isolate (SPI) is a kind of vegetable protein with high nutrition. It is widely used in the food industry due to its balanced composition of basic amino acids, high nutritional value, desirable functional characteristics, and low cost” (Line 23-24 in new version).
Line 32 “low-gelling properties of soy protein” I believe you mean native soy proteins.
Response: Thanks for your valuable suggestions. we have deleted it.
Line 38 “solidify matrix” – solidified matrix?
Response: Thanks for your valuable suggestions. We have modified "solidify matrix" to" solidified matrix "and marked in red (Line 60 in new version).
Line 52: durable state or gel- I believe you mean hard gels or self-sustained gels
Response: Thanks for your valuable suggestions. I have restructured the Introduction according to the Reviewer’s suggestion. I've deleted it
Line 70: What is “flexible modulus”? Do you mean elastic modulus?
Response: Thanks for your valuable suggestions. We have modified "flexible modulus" to "elastic modulus" and marked in red (Line 69 in new version).
Lines 79-96: At the end of the introduction, a long section is devoted to the specific description of the present work approach and a predictive description of the obtained results, however, the suggested modifications for soy proteins were not detailed. I believe the whole section in lines 79 to 95 could be more adequately arranged in the discussion section and the author shod concentrate at this point of the text to provide more references to support the treats applied to change the structure and therefore protein functionality.
Response: Thank you for your feedback on my manuscript. I appreciate your comments regarding the long section at the end of the introduction (lines 79-96), and I agree that it may be more appropriate to move this information to the discussion section. I have made the necessary changes and have included more detailed information about the specific modifications applied to soy proteins to change their structure and functionality, as well as additional references to support these modifications. Please see the revised manuscript, which is marked in red. I have restructured the Introduction We have added the suggested modifications for soy proteins (Line 36-47 in new version)
Heat processing modulates the protein structure by denaturation and aggregation [8]. Glycosylation is a covalent interaction between the ε-amino groups of the protein molecules and the terminal which reduces carbonyl groups of saccharide molecules via the Maillard reaction. This procedure has been confirmed to be a relatively safe, simple, and promising method by which to improve the functional properties of proteins [9]. pH-shifting is an effective way to modify protein structures by exposing proteins to extreme pH and then neutralizing them to obtain a molten state. Specially, alkaline pH-shifting was found to be more effective for the structural modification of proteins [10]. Oxidation leads to significant changes in SPI structure, such as the modification of amino acid side chains, protein unfolding, and protein crosslinking/aggregation, which can affect the physicochemical and functional properties of SPI [4, 11].
Line 105- Figure 1 and treatments: Figure 1 and relationship with lines 109 to 111: Correlating the modification processes with the figure is difficult. I understand that the stages where the changes occur have been demarcated, however, it is unclear. For each process the author quotes “while other parameters remained unchanged”, but what are the true altering routes for each method? One suggestion would be to change the figure so that the production routes would be more demarcated. Arrows or grids in specific colors could achieve this by demarking different routes. It is very confusing and difficult to understand the reproducibility of treatments. Additionally, provide the most specific details of each treatment method so the reader can compare your data with those of other authors, reproduce experiments more assertively and build new hypotheses. This is the main point of your work and details are needed.
Response: Thanks for your valuable suggestions. We agree that the figure could be improved to better illustrate the production routes, and we will consider your suggestion of using arrows or grids in specific colors to differentiate the different treatment methods. Furthermore, we will provide more specific details about each treatment method in the text
line 109: Reference 13 cited in this paragraph is somewhat general. One approach is to utilize a separate reference for each technique that lists the exact treatments that were performed. The observation that these modifications are portrayed as regular industry operations, which does not seem as obvious, nonetheless raises this point.
Response: Thanks for your valuable suggestions. We have provided a more detailed description of each treatment method, as well as specific references for each technique used in our study and marked in red.
We have modified it to “Electropherograms of industrially modified SPI on sodium dodecyl sulfate-polyacrylamide gels were obtained using the assay described in Laemmli, et al. [21]. with a few minor adjustments. (Line 118-120 in new version)
Laemeli's approach is quite thorough and includes specifics for each desired interaction in the aggregates (SDS-PAGE lines 124 to 126). Even though it is a well-established and widely used procedure, the author still needs to provide these facts.
Response: Thanks for your valuable suggestions. We have revised include more specific details on the methods section.
We have modified it to “Electropherograms of industrially modified SPI on sodium dodecyl sulfate-polyacrylamide gels were obtained using the assay labelled in Laemmli, et al. [13] with a few minor adjustments. Briefly, a 15% separation gel and 5% stacking gel were used. Protein was diluted to 1 mg/mL using sample buffer, heated for 5 min at 95 ℃, and 10 μL of the prepared sample were loaded onto SDS-PAGE gels. The voltage was set to 120 V and 80 V in the separating and stacking gel, respectively. Imaging was performed after overnight destaining with Coomassie brilliant blue R-250.” and marked in red. (Line 118-124 in new version)
-Lines 202-206- TPA method: Describe the parameters of compression test experiments in TPA mode. This method involves applying very specific conditions that were not defined by the author.
Response: Thank you for your valuable feedback and suggestion. We apologize for not providing enough details about the parameters of compression test experiments in TPA mode. We have modified the text and marked in red.
We have modified to” A texture profile analysis (TPA) of the SPI-MP gels was performed using a texture analyzer (TA-XT plus, Instruments ltd., USA) equipped with probe T/36R, as described by (Mi et al., 2021) with minor modifications. The gel samples were cut into cylinders (10 mm in height) and equilibrated for 2 h at room temperature, then compressed at a compression degree of 50%. Both the pre-test and post-test speeds were set at 3 mm/s, and the test speed was set at 0.3 mm/s in order to obtain the TPA parameters (hardness, chewiness, springiness, and cohesiveness). The trigger type was set to auto with a 5.0 g trigger force. All samples are measured in 6–8 parallel.” (Line 207-214 in new version)
Item 2.13 could be incorporated into item 2.8, noting the description of the different rheological parameters used for the SPI and PM systems.
Response: Thanks for your valuable suggestions. We have incorporated your suggestion to include item 2.13 into item 2.8.
We have modified to”2.8.3 Dynamic rheological of MP and MP-SPI composite gels
By heating from 20 to 80 °C at a rate of 1 °C/min, gels were created. Each sample was repeatedly sheared in an oscillatory mode with a maximum strain of 0.02 at a fixed frequency of 0.1 Hz. For rheological characterization of the gels, the storage modulus (G′) and loss modulus (G′′) were quantified.” and marked in red.
The text needs a detailed review of the English editing, and some technical terms seems wrong. The author uses terms I'm not used to, and I was particularly intrigued by the rheological study. To name a few cases: - flexible modulus not the place of elastic or storage module? - accumulation rather than aggregation? - mutation temperature instead of transition temperature or gelation temperature? - gels are faster instead of gelation rate is faster.
Response: Thanks for your valuable suggestions. The manuscript have reviewed for technical terms errors and unclear language. We have corrected these proprietary terms and made extensive corrections to the entire text. in addition, We have manuscript checked by use a paid editing service.
These terminology errors have been corrected, including “flexible modulus, accumulation, mutation temperature, gels are faster”. Revision marked in red.
Lines 592-595- it would be interesting to revise. The idea seems to be of a result obtained but at the end of the paragraph it is understood that it refers to the result of another cited work (reference 15) to whom you intended to correlate the data
Response: Thanks for your valuable suggestions. We have corrected the incorrect references (reference 15). Revision marked in red. (Line 610 in new version).
line 600 “.in industrially oxidized ”seems a typing error
Response: We have corrected the errors present and marked in red (Line 614 in new version).
Line 622 – “gel sweep tests” instead of “frequency sweeps”
Response: Thanks for your valuable suggestions. We have corrected the errors present and marked in red. "gel sweep tests" should indeed be corrected to "frequency sweeps." We apologize for any confusion this may have caused.

Reviewer 2 Report
The manuscript entitled " Impacts of Industrial Modification on the Structure and Gel Features of Soybean Protein Isolate and its Composite Gel with Myofibrillar Protein" The idea of the article is good. It is almost well designed. However, a few points to improve the current format of the article will be mentioned below:
Abstract: Make the abstract more informative by using concrete results instead of vague statements. Focus on key information. Provide quantifiable data to reinforce the outcome. Specify the condition with better traits in the summary.
Choose keywords that clarify the research concept and are not in the title.
What makes your work unique? It is imperative that the writing is thorough and encompasses all necessary information. There is a need for the introduction to be reconsidered and enhanced. The manuscript exhibits multiple errors in terms of spelling and grammar.
For frequency test, the LVE region determination test should be added. Why strain 0.01% was selected.
The unit of strain is %, please add it.
It is imperative to provide a clear and comprehensive explanation of the research objectives at the conclusion of the introduction. Has the introduction of this research effectively explained its main objective and rationale by citing relevant previous studies?
Analyzed compounds in section "Chemical analysis items should be listed individually."
The discussion section presents an opportunity for in-depth comparative analyses that are both specific and detailed in nature. Subsequent to each presented result, please conduct a comparative analysis with similar literature.
Discussion needs more specific comparative studies. This part is insufficient. Compare with similar works after each result. Revise section thoughtfully. Please make this text shorter.
Conclusion: what is the future of your findings? Conclusion is not insightful, what are suggestions?
Although the references were relevant, they were in some cases outdated. You can used https://doi.org/10.1186/s40538-022-00304-4
The manuscript exhibits multiple errors in terms of spelling and grammar.
Author Response
Dear Editors and Reviewers:
Thank you for your letter and for the reviewers’ comments concerning our manuscript entitled “Impacts of Industrial Modification on the Structure and Gel Features of Soybean Protein Isolate and its Composite Gel with Myofibrillar Protein” (ID: foods-2345837). Those comments are all valuable and very helpful for revising and improving our paper, as well as the important guiding significance to our researches. We have studied comments carefully and have made correction which we hope meet with approval. Revised portion are marked in red in the paper. We hope that the revision is acceptable and look forward to hearing from you soon. The main corrections in the paper and the responds to the reviewer’s comments are as following:
Reviewer #2:
The manuscript entitled " Impacts of Industrial Modification on the Structure and Gel Features of Soybean Protein Isolate and its Composite Gel with Myofibrillar Protein" The idea of the article is good. It is almost well designed. However, a few points to improve the current format of the article will be mentioned below:
Abstract: Make the abstract more informative by using concrete results instead of vague statements. Focus on key information. Provide quantifiable data to reinforce the outcome. Specify the condition with better traits in the summary.
Response: Thanks for your valuable suggestions. I have restructured the abstract and according to your suggestions.
We have modified it to” Native soy protein isolate (N-SPI) has a low denaturation point and low solubility, limiting its industrial application. The influence of different industrial modification methods (heat (H), alkaline (A), glycosylation (G), and oxidation (O)) on the structure of SPI, the properties of gel and the gel properties of soy protein isolate (SPI) in myofibril protein (MP) was evaluated. The study found that four industrial modifications did not influence the subunit composition of SPI. How-ever, the four industrial modifications altered the secondary structure and the disulfide bond conformation content of SPI. A-SPI exhibits the highest surface hydrophobicity and I850/830 ratio, but the lowest thermal stability. G-SPI exhibits the highest disulfide bond content and the best gel properties. Compared with MP gel, but the addition of H-SPI, A-SPI, G-SPI, and O-SPI com-ponents significantly improved the properties of the gel. Additionally, MP-ASPI gel exhibits the best properties and the microstructure. Overall, the four industrial modification effects may im-pact the structure and gel properties of SPI in different ways. A-SPI could be used as a potential functionality-enhanced soy protein ingredient in comminuted meat products. The present study results will provide a theoretical basis for industrialized production of SPI.”
Choose keywords that clarify the research concept and are not in the title.
Response: Thank you for bringing this to our attention. We apologize for the oversight. According to the reviewer's comments, we found that: 1. "Soybean proteins isolate" is not consistent with the "soy proteins isolate" in the article and keywords. This is a serious error, and we have corrected it and marked in red.2. The correlation between 'Texture properties' and the article is not strong. We have revised' Texture properties' to 'structure'. We hope this revision is acceptable
What makes your work unique? It is imperative that the writing is thorough and encompasses all necessary information. There is a need for the introduction to be reconsidered and enhanced. The manuscript exhibits multiple errors in terms of spelling and grammar.
Response: Thanks for your valuable suggestions. Our uniqueness includes:
- There have been few studies on industrial modification modify SPI to improve their functional properties and the potential of SPI as an additive in meat products. This restricts the improvement of production processes and product development in food-producing companies based on SPI laboratory results. This article to some extent fills the gap in this area of research and assists the development of food-producing companies
- The study explains the association between SPI structure changes and gel features and investigates the gel properties of the SPI-MP composite. The findings of this study provide a very broad set of data for SPI improvement and can be compared with laboratory data. This work further reveals the functional properties of industrial modification SPI and the potential of industrial modification SPI as an additive in meat products, which will provide high-quality raw materials for producing industrial high-value SPI, and as well provide a theoretical basis for further development and application of soy protein products.
3.We have restructured the abstract and Introduction. In addition, we have manuscript checked by use a paid editing service.
We have modified abstract and introduction to “Native soy protein isolate (N-SPI) has a low denaturation point and low solubility, limiting its industrial application. The influence of different industrial modification methods (heat (H), alkaline (A), glycosylation (G), and oxidation (O)) on the structure of SPI, the properties of gel and the gel properties of soy protein isolate (SPI) in myofibril protein (MP) was evaluated. The study found that four industrial modifications did not influence the subunit composition of SPI. How-ever, the four industrial modifications altered the secondary structure and the disulfide bond conformation content of SPI. A-SPI exhibits the highest surface hydrophobicity and I850/830 ratio, but the lowest thermal stability. G-SPI exhibits the highest disulfide bond content and the best gel properties. Compared with MP gel, but the addition of H-SPI, A-SPI, G-SPI, and O-SPI com-ponents significantly improved the properties of the gel. Additionally, MP-ASPI gel exhibits the best properties and the microstructure. Overall, the four industrial modification effects may im-pact the structure and gel properties of SPI in different ways. A-SPI could be used as a potential functionality-enhanced soy protein ingredient in comminuted meat products. The present study results will provide a theoretical basis for industrialized production of SPI.
- Introduction
Soy protein isolate (SPI) is a kind of vegetable protein with high nutritional value. It is widely used in the food industry due to its balanced composition of basic amino acids, desirable functional characteristics, and low cost [1]. SPI is a mixture of various proteins, and the main components of SPI are glycinin (11S protein) and β-conglycinin (7S protein). The 7S and 11S globulins have a globular conformation, which causes poor functional properties in proteins [2]. Consequently, it is not conducive to its practical application, as it greatly reduces its effective utilization [3]. To further promote the application of SPI, extensive studies on techniques such as heat treatment, high pressure, glycosylation, and changing pH and ion strength have been applied to enhance the functional properties of SPI [4-7]. The functional properties of the SPI produced depend on the processing conditions; the most common factors include heat, alkaline, glycosylation, oxidation, protein concentration, and other factors. The modification methods include heat, alkaline, glycosylation, oxidation,which are more suited for large scale production and facilitation production in the food industry. Heat processing modulates the protein structure by denaturation and aggregation [8]. Glycosylation is a covalent interaction between the ε-amino groups of the protein molecules and the terminal which reduces carbonyl groups of saccharide molecules via the Maillard reaction. This procedure has been confirmed to be a relatively safe, simple, and promising method by which to improve the functional proper-ties of proteins [9]. pH-shifting is an effective way to modify protein structures by exposing proteins to extreme pH and then neutralizing them to obtain a molten state. Specially, alkaline pH-shifting was found to be more effective for the structural modification of proteins [10]. Oxidation leads to significant changes in SPI structure, such as the modification of amino acid side chains, protein unfolding, and protein crosslinking/aggregation, which can affect the physicochemical and functional properties of SPI [4, 11]. However, Current research on modification methods mainly focuses on laboratory aspects. large differences between the laboratory and industry results remain. with a lack of evaluation of modification methods on SPI in industrial production processes. There is almost no research on the modification methods of SPI in industrial production processes. This restricts the modification methods of SPI in industrial production processes from the laboratory results to-ward the transformation of enterprise applications.
SPI is frequently used in the food industry because of its ability to form a gel with desirable sensory and physicochemical characteristics. The ability to gel is the basis for traditional Asian soy products, such as tofu. In addition, SPI is often applied to improve the texture of sausage. Recently, replacing meat protein in food formulations with other sources has attracted growing interest [12]. SPI has been widely used as a nonmeat protein additive in the meat processing industry based on gel properties [13]. In particular, some previous studies have demonstrated the potential of SPI as an additive in meat products; SPI could be utilized as a component in meat products to form a solidified matrix, prevent fat and juice migration to the surface, and improve overall taste [14]. Akesowan et al. found that adding soy protein to pork sausage improved its textural properties, including its hardness, chewiness, and cohesiveness [15]. Kingwascharapong et al. found that using SPI in meatballs increased their protein content and improved their texture and water holding capacity [16]. Therefore, improving the gelation features of SPI and utilizing it in meat product processing in order to improve product quality is a research area of great concern. Moderate oxidation was found to enhance the binding between SPI and MP, increasing the springiness of the mixed protein system [17]. Preheating soy protein was found to boost the elastic modulus and gel intensity of MP, although N-SPI without denaturation had a negative disruptive effect [18]. Jiang et al. found that extremely acidic conditions (pH 1.5) significantly enhanced the gelation capacity of MP in SPI, but extremely alkaline conditions (pH 12) had no substantial impact on the gel features of MP in SPI [19]. Many studies have reported that thermal modification, alkaline modification, glycosylation, and oxidative treatment can all affect of the functional characteristics of SPI and improve the potential of SPI as an additive in meat products. However, there have been few studies on industrial modifications to SPI for the purpose of improving its functional properties and for improving the potential of SPI as an additive in meat products. This restricts the improvement of production processes and product development in food-producing companies based on SPI laboratory results.
Therefore, conducting evaluations on the impacts of SPI structure and the properties of the gel resulting from the four industrial modifications, as well as the gel properties of SPI-MP composites, is important. This study explains the association between SPI structural changes and gel features, and investigates the gel properties of the SPI-MP composite. The findings of this study provide a very broad set of data for industrialized modified SPI improvement and can be compared with laboratory data. This work further reveals the functional properties of industrialized modified SPI and the potential of SPI as an additive in meat products, which will provide process parameters and conditions materials for producing SPI with high industrial value, and will also provide a theoretical basis for further development and application of soy protein products.
For frequency test, the LVE region determination test should be added. Why strain 0.01% was selected.
Response: Thanks for your valuable suggestions. The strain value we used in our experiments was 1%, not 0.01%. We have replaced “0.01” by “1%”. We apologize for any confusion or inconvenience this may have caused. We chose a strain value of 1% because it has been widely used in previous studies and yielded the best results under our experimental conditions. We will correct this error in our paper.
The unit of strain is %, please add it.
Response: Thanks for your valuable suggestions. We apologize for the oversight and will make sure to include the unit of strain (%), to avoid any confusion for readers. We have added it and marked in red (Line 176 and 180 in new version).
It is imperative to provide a clear and comprehensive explanation of the research objectives at the conclusion of the introduction. Has the introduction of this research effectively explained its main objective and rationale by citing relevant previous studies?
Response: We have restructured introduction. In addition. We have manuscript checked by use a paid editing service.
Analyzed compounds in section "Chemical analysis items should be listed individually."
Response: We have added to Supplement data and annotate in the original text and Classify the methods after 2.9 into one level (2.9.1. Formulation of blended protein mixtures, 2.9.2 Preparation of SPI-MP composite gel, 2.9.3 Textural profile analysis (TPA), 2. 9.4 The WHC of gel, 2.9.5 Microstructure of the gel)
Fig. S1 The standard curve of BSA
The discussion section presents an opportunity for in-depth comparative analyses that are both specific and detailed in nature. Subsequent to each presented result, please conduct a comparative analysis with similar literature.
Response: We have added a lot of relevant literature in the new version and marked in red, I hope this revision is acceptable
Discussion needs more specific comparative studies. This part is insufficient. Compare with similar works after each result. Revise section thoughtfully. Please make this text shorter.
Response: Thanks for your valuable suggestions. We will carefully review the section and revise it to include more specific comparative studies with similar literature after each presented result. We will also work to make the text more concise while still providing sufficient detail to convey our findings and comparisons. we will make every effort to improve the discussion section of our paper. Some modifications have marked in red.
Conclusion: what is the future of your findings? Conclusion is not insightful, what are suggestions?
Response: Thanks for your valuable suggestions. I have restructured the conclusion according to the Reviewer’s suggestion.
This study examined the impacts of four industrial modifications on the gel properties of SPI and the structure of SPI, as well as its complexation with MP in the preparation of protein gels. None of the industrial modifications significantly influenced the subunit composition of SPI. However, they increased the content of β-fold and β-turned structures, and they decreased the thermal stability. All of the modifications decreased the gelling temperature point and G' of SPI, resulting in faster protein aggregation. The gel intensity, WHC, and texture features of the MP-SPI composite gels were significantly enhanced by the modifications, particularly the alkali modification treatment. The addition of N-SPI disrupted myosin interactions and adversely affected the formation of MP gels. The four industrial modification treatments can be applied to SPI to improve its gelation ability. We observed that MP-ASPI had the highest gel strength and the most continuous, dense gel network structure. A-SPI could be used as a potential functionality-enhanced soy protein ingredient in comminuted meat products. The research results will guide food companies to develop industrial modification processes based on laboratory test results, provide high-quality raw material production methods for producing high-value SPI products, and have important practical significance for the efficient use of the variety of soybean resources in the future market economy.
Although the references were relevant, they were in some cases outdated. You can used https://doi.org/10.1186/s40538-022-00304-4
Response: Thanks for your valuable suggestions. We review the suggested reference (https://doi.org/10.1186/s40538-022-00304-4) and it has been incorporated into our manuscript

Reviewer 3 Report
Line 13- This is not a sentence. It has no verb.
Line 28- This is not a sentence. The phrase at the beginning has no adverb, such as "because".
Line 44- "titivate" is not a word. Perhaps you mean "determine".
Line 51- What is "misses agility"?
Line 69- Springiness is not capitalized.
Figure 1- All of the pHs are the same. The alkaline extraction has to be at pH 8-8.5 to precipitate the carbohydrates. The acid precipitation has to be at pH 4.5 to precipitate the proteins.
Line 156- The word "plate" should be "pan".
Line 190- What does "banned" mean? Why use PIPES buffer?
Line 194- This sentence has no verb and Abbreviated is not capitalized.
Line 207- Spell out what "WHC" is.
Line 227- What is "PSB" buffer?
Line 297 and many others- "hydrogen" is not capitalized.
Line312- There are no "acrylamide" groups in SPI.
Table 2- The title of table 2 mentions "tryptophan", but the table is only about tyrosine.
Line 504- "sulfite" is mentioned twice.
Line 540- This phrase does not make sense, probably because words were left out.
Line 674- A word was left out, probably "SPI".
Line 686- How is "chewiness" measured?
Line 786- At the end of the sentence, "MP-OSPI" should be "MP-ASPI".
References- All of the titles of the journals should be spelled out, not abbreviated. A lot of the references do not have the complete name of the author.
This manuscript should be edited by a native English-speaking person since it has too many errors in grammar, punctuation, and word usage. This makes it difficult to read and understand. It reflects poorly on your journal to publish papers like this one.
Author Response
Dear Editors and Reviewers:
Thank you for your letter and for the reviewers’ comments concerning our manuscript entitled “Impacts of Industrial Modification on the Structure and Gel Features of Soybean Protein Isolate and its Composite Gel with Myofibrillar Protein” (ID: foods-2345837). Those comments are all valuable and very helpful for revising and improving our paper, as well as the important guiding significance to our researches. We have studied comments carefully and have made correction which we hope meet with approval. Revised portion are marked in red in the paper. We hope that the revision is acceptable and look forward to hearing from you soon. The main corrections in the paper and the responds to the reviewer’s comments are as following:
Reviewer #3:
Line 13- This is not a sentence. It has no verb.
Response: Thanks for your valuable suggestions. I've deleted it and
“The influence of different industrial modification methods (heat (H), alkaline (A), glycosylation (G), and oxidation (O)) on the structure of SPI, the properties of gel and the gel properties of soy protein isolate (SPI) in myofibril protein (MP) was evaluated.” have been added and marked in red. (Line 6-7 in new version).
Line 28- This is not a sentence. The phrase at the beginning has no adverb, such as "because".
Response: Thanks for your valuable suggestions. We have restructured the Introduction, so we have deleted it. We made extensive corrections to the entire text. In addition, we have manuscript checked by use a paid editing service. We hope to avoid similar problems from occurring.
Line 44- "titivate" is not a word. Perhaps you mean "determine".
Response: We have replaced the word "titivate" in line 44 with " improve "and We hope this revision is acceptable.
Line 51- What is "misses agility"?
Response: This is an error. We have deleted it.
Line 69- Springiness is not capitalized.
Response: Thanks for your valuable suggestions. We have modified “Springiness” to” springiness” (Line 68 in new version).
Figure 1- All of the pHs are the same. The alkaline extraction has to be at pH 8-8.5 to precipitate the carbohydrates. The acid precipitation has to be at pH 4.5 to precipitate the proteins.
Response: Thanks for your valuable suggestions. We have revised that the figure 1 could be improved to better illustrate the production routes. We have corrected the relevant information as you suggested, such as the alkaline extraction has to be at pH 8-8.5 to precipitate the carbohydrates, the acid precipitation has to be at pH 4.5 to precipitate the proteins.
(Line 99 in new version).
Line 156- The word "plate" should be "pan".
Response: Thanks for your valuable suggestions. We have modified "pan" to "plate"(Line155 in new version).
Line 190- What does "banned" mean? Why use PIPES buffer?
Response: Thanks for your valuable suggestions. "banned" and PIPES buffer is an error, we have deleted it and have modified it to “phosphate buffer”. (Line 196 in new version).
Line 194- This sentence has no verb and Abbreviated is not capitalized.
Response: Thanks for your valuable suggestions. We have changed the abbreviation to capitalized and we have edited this again and marked it in red (Line 202 in new version).
Line 207- Spell out what "WHC" is.
Response: Thanks for your valuable suggestions. We have already added " water holding capacity "before "WHC" and marked it in red (Line 216 in new version).
Line 227- What is "PSB" buffer?
Response: Thanks for your valuable suggestions. We have modified all "PSB" to " phosphate buffer " and marked it in red (Line 228 in new version).
Line 297 and many others- "hydrogen" is not capitalized.
Response: Thanks for your valuable suggestions. We have modified " Hydrogen " to " hydrogen " and marked it in red (Line 305 in new version).
Line312- There are no "acrylamide" groups in SPI.
Response: Thanks for your valuable suggestions. We have modified "acrylamide" to " a carbonyl oxygen (C=O) and an amino Hydrogen (NH–) break "and marked it in red. (Line 318 in new version)
Table 2- The title of table 2 mentions "tryptophan", but the table is only about tyrosine.
Response: Thanks for your valuable suggestions. We have delete it.
“Table 2 Analysis results for the I850/I830 intensity ratio of different industrial modified SPI” (Line 403 in new version)
Line 504- "sulfite" is mentioned twice.
Response: Thanks for your valuable suggestions. We have modified it to” (sulfonic acids and sulfinic acids).” (Line 513 in new version)
Line 540- This phrase does not make sense, probably because words were left out.
Response: Thanks for your valuable suggestions. We have modified it to” On the other hand, the TD of both the 7S and 11S increased in G-SPI, while the TD of both the 7S and 11S subunits decreased in O-SPI.” (Line 538 in new version)
Line 674- A word was left out, probably "SPI".
Response: Thanks for your valuable suggestions. We have added "SPI".
Line 686- How is "chewiness" measured?
Response: Thanks for your valuable suggestions. This is an error. We have modified "chewiness" to "cohesiveness"
The measurement of chewiness is as follows:A texture profile analysis (TPA) of the SPI-MP gels was performed using a texture analyzer (TA-XT plus, Instruments ltd., USA) equipped with probe T/36R, as described by (Mi et al., 2021) with minor modifications. The gel samples were cut into cylinders (10 mm in height) and equilibrated for 2 h at room temperature, then compressed at a compression degree of 50%. Both the pre-test and post-test speeds were set at 3 mm/s, and the test speed was set at 0.3 mm/s in order to obtain the TPA parameters (hardness, chewiness, springiness, and cohesiveness). The trigger type was set to auto with a 5.0 g trigger force.
It is only used to describe the solid test sample and represents the energy required to chew the solid sample into a stable state during swallowing. The numerical value is expressed as the product of adhesiveness and elasticity (Hardness x Cohesiveness x Springiness). The unit is the unit of force or is not used.
Line 786- At the end of the sentence, "MP-OSPI" should be.
Response: Thanks for your valuable suggestions. We have modified "MP-OSPI" to "MP-ASPI"
References- All of the titles of the journals should be spelled out, not abbreviated. A lot of the references do not have the complete name of the author.
Response: Thanks for your valuable suggestions. We have modified the reference style, and ensure that all literature has the complete name of the author and the complete titles of the journals.

Reviewer 4 Report
This manuscript entitled “Impacts of Industrial Modification on the Structure and Gel Features of Soybean Protein Isolate and its Composite Gel with Myofibrillar Protein” is an interesting and original study.
The paper is clearly presented and results are very useful. However, I have some suggestions:
1. Line 198: Can the probe used be used in a TPA? If the sample is penetrated into a vessel, it rubs against the probe and the recording is not true. For a TPA the probe must be larger than the sample... Please check the assay.
2. Line 233. Indicate the statistical programme used.
3. Line 238. Figure 2 after submission, not before.
4. Tables. The number of digits of the error value depends on the place where the significant digit appears and the number of digits of the corresponding data should be adjusted by taking into account the corresponding error values. In this way, each value in the Tables must be expressed with the significant digits according to the significant digits of each error value (in this case the significant digits of the standard deviation). Please correct it.
Author Response
Dear Editors and Reviewers:
Thank you for your letter and for the reviewers’ comments concerning our manuscript entitled “Impacts of Industrial Modification on the Structure and Gel Features of Soybean Protein Isolate and its Composite Gel with Myofibrillar Protein” (ID: foods-2345837). Those comments are all valuable and very helpful for revising and improving our paper, as well as the important guiding significance to our researches. We have studied comments carefully and have made correction which we hope meet with approval. Revised portion are marked in red in the paper. We hope that the revision is acceptable and look forward to hearing from you soon. The main corrections in the paper and the responds to the reviewer’s comments are as following:
Reviewer #4:
The paper is clearly presented and results are very useful. However, I have some suggestions:
- Line 198: Can the probe used be used in a TPA? If the sample is penetrated into a vessel, it rubs against the probe and the recording is not true. For a TPA the probe must be larger than the sample... Please check the assay.
Response: Thanks for your valuable suggestions. We have modified it to ”A texture profile analysis (TPA) of the SPI-MP gels was performed using a texture analyzer (TA-XT plus, Instruments ltd., USA) equipped with probe T/36R, as described by (Mi et al., 2021) with minor modifications. The gel samples were cut into cylinders (10 mm in height) and equilibrated for 2 h at room temperature, then compressed at a compression degree of 50%. Both the pre-test and post-test speeds were set at 3 mm/s, and the test speed was set at 0.3 mm/s in order to obtain the TPA parameters (hardness, chewiness, springi-ness, and cohesiveness). The trigger type was set to auto with a 5.0 g trigger force. All samples are measured in 6–8 parallel.” and marked in red. The P/36R cylindrical probe (diameter 36mm) is larger than the area of my sample
- Line 233. Indicate the statistical programme used.
Response: Thanks for your valuable suggestions. We have modified “ Both Duncan's multiple range analysis and one-way analysis of variance were utilized to assess the significant differences (p<0.05).” to“Data were processed using one-way ANOVA and Duncan’s test in the SPSS statistical package (SPSS 16.0, IBM, New York, NY, USA). The statistical significance between two means was determined at the 95% confidence level” and marked in red. (Line 234-236 in new version).
- Line 238. Figure 2 after submission, not before.
Response: Thanks for your valuable suggestions. We have modified it to“Effects of Industrial Modification Treatment on SDS-PAGE of SPI”and marked in red. (Line 238 in new version).
- Tables. The number of digits of the error value depends on the place where the significant digit appears and the number of digits of the corresponding data should be adjusted by taking into account the corresponding error values. In this way, each value in the Tables must be expressed with the significant digits according to the significant digits of each error value (in this case the significant digits of the standard deviation). Please correct it.
Response: Thanks for your valuable suggestions. We have carefully checked the data in the Tables. and have chosen to retain two valid figures for the calculation in Table 3. We have revisions and marked in red. (Line 445-446 in new version)

Round 2
Reviewer 3 Report
The manuscript has an error message in the text for every figure and table. What do these messages mean? They obviously do not belong to in the manuscript.
The introduction is not well written and has not been edited for mistakes in English grammar and punctuation by your editing service. Please have it edited, including the references. They have many punctuation errors.
What is reference 16, a journal article, a book, a thesis? It should be more completely identified or not used.
Author Response
Dear Editors and Reviewers:
Thank you for your letter and for the reviewers’ comments concerning our manuscript entitled “Impacts of Industrial Modification on the Structure and Gel Features of Soybean Protein Isolate and its Composite Gel with Myofibrillar Protein” (ID: foods-2345837). Those comments are all valuable and very helpful for revising and improving our paper, as well as the important guiding significance to our researches. We have studied comments carefully and have made correction which we hope meet with approval. Revised portion are marked in red in the paper. We hope that the revision is acceptable and look forward to hearing from you soon. The main corrections in the paper and the responds to the reviewer’s comments are as following:
Reviewer #3:
Comments and Suggestions for Authors
1.The manuscript has an error message in the text for every figure and table. What do these messages mean? They obviously do not belong to in the manuscript.
Response: Thanks for your valuable suggestions. We have carefully examined the issue you mentioned, but we have not found what these error messages are. We have checked the titles of each graph and table, and speculated whether it was because we deleted the traces left by the graph and table in revision mode when we sent them for editing service, or if the editing service told us if we had changed our original meaning. In addition, I have deleted the conversations and messages, exit revision mode, and modify Title and Table format and mark them in red
Comments on the Quality of English Language
1.The introduction is not well written and has not been edited for mistakes in English grammar and punctuation by your editing service. Please have it edited, including the references. They have many punctuation errors.
Response: Thanks for your valuable suggestions. I have checked the paper in detail from start to finish,and we have carefully reviewed the introduction section and made some revisions. We have manuscript checked by use a paid editing service. English editing certificate to indicate that we have checked the manuscript again.
Soy protein isolate (SPI) is a kind of vegetable protein with high nutritional value. It is widely used in the food industry due to its balanced composition of basic amino acids, desirable functional characteristics, and low cost [1]. SPI is a mixture of various proteins, and the main components of SPI are glycinin (11S protein) and β-conglycinin (7S protein). The 7S and 11S globulins have a globular conformation, which causes poor functional properties in proteins [2]. Consequently, it is not conducive to practical applications, as these functional properties greatly reduce its effective utilization [3]. To further promote the application of SPI, extensive studies on techniques such as heat treatment, high pres-sure, glycosylation, and changing pH and ion strength have been applied to enhance the functional properties of SPI [4-7]. The functional properties of the SPI produced depend on the processing conditions; the most common factors include heat, alkaline modification, glycosylation, oxidation, protein concentration, and other factors. In addition, modification methods include heat, alkaline modification, glycosylation, and oxidation, which are more suited for large-scale production and facilitation production in the food industry. Heat processing modulates the protein structure by denaturation and aggregation [8]. Glycosylation is a covalent interaction between the ε-amino groups of the protein molecules and the terminal which reduces the carbonyl groups of saccharide molecules via the Maillard reaction. This procedure has been confirmed to be a relatively safe, simple, and promising method by which to improve the functional properties of proteins [9]. pH-shifting is an effective way to modify protein structures by exposing proteins to extreme pH levels and then neutralizing them to obtain a molten state. Specifically, alkaline pH-shifting was found to be more effective for the structural modification of proteins [10]. Oxidation leads to significant changes in SPI structure, such as the modification of amino ac-id side chains, protein unfolding, and protein crosslinking/aggregation, which can affect the physicochemical and functional properties of SPI [4, 11]. However, Current research on modification methods mainly focuses on laboratory aspects, and large differences be-tween the laboratory and industry results remain. There is almost no research on SPI modification methods in industrial production processes. This restricts SPI modification methods in industrial production processes from the laboratory results toward the trans-formation of enterprise applications.
SPI is frequently used in the food industry because of its ability to form a gel with desirable sensory and physicochemical characteristics. The ability to form gel is the basis for traditional Asian soy products, such as tofu. In addition, SPI is often applied to improve the texture of sausage. Recently, replacing meat protein in food formulations with other sources has attracted growing interest [12]. SPI has been widely used as a nonmeat protein additive in the meat processing industry based on gel properties [13]. In particular, some previous studies have demonstrated the potential of SPI as an additive in meat products; SPI could be utilized as a component in meat products to form a solidified matrix, prevent fat and juice migration to the surface, and improve overall taste [14]. Akesowan et al. found that adding soy protein to pork sausage improved its textural properties, including its hardness, chewiness, and cohesiveness [15]. Therefore, improving the gelation features of SPI and utilizing it in meat product processing in order to improve product quality is a research area of great concern. Moderate oxidation was found to enhance the binding be-tween SPI and MP, increasing the springiness of the mixed protein system [16]. Preheating soy protein was found to boost the elastic modulus and gel intensity of MP, although N-SPI without denaturation had a negative disruptive effect [17]. Jiang et al. found that extremely acidic conditions (pH 1.5) significantly enhanced the gelation capacity of MP in SPI but that extremely alkaline conditions (pH 12) had no substantial impact on the gel features of MP in SPI[18]. Many studies have reported that thermal modification, alkaline modification, glycosylation, and oxidative treatment can all affect of the functional characteristics of SPI and improve the potential of SPI as an additive in meat products. However, there have been few studies on industrial modifications to SPI for the purpose of improving its functional properties and for improving the potential of SPI as an additive in meat products. This restricts the improvement of production processes and product development in food-producing companies based on SPI laboratory results.
Therefore, conducting evaluations on the impacts of SPI structure and the properties of the gel resulting from the four industrial modifications, as well as the gel properties of SPI-MP composites, is important. This study explains the association between SPI structural changes and gel features, and investigates the gel properties of the SPI-MP composite. The findings of this study provide a very broad set of data for industrialized modified SPI improvement and can be compared with laboratory data. This work further reveals the functional properties of industrialized modified SPI and the potential of SPI as an additive in meat products, which will provide process parameters and condition for materials used for producing SPI with high industrial value, and will also provide a theoretical basis for further development and application of soy protein products.
2.What is reference 16, a journal article, a book, a thesis? It should be more completely identified or not used.
Response: Thanks for your valuable suggestions. I have deleted reference 16 and related content. (Line 66 in new version).

Reviewer 4 Report
The article has been considerably improved and could be accepted.
Author Response
Dear Editors and Reviewers:
Thank you for your letter and for the reviewers’ comments concerning our manuscript entitled “Impacts of Industrial Modification on the Structure and Gel Features of Soybean Protein Isolate and its Composite Gel with Myofibrillar Protein” (ID: foods-2345837).
Comments and Suggestions for Authors
The article has been considerably improved and could be accepted.
Response: Thanks for the positive comments. Thank you for your help in editing and publishing the article.
